

# Branes, quivers and wave-functions

**Taro Kimura[1], Miłosz Panfil[2*], Yuji Sugimoto[3] and Piotr Sułkowski[2,4]**

**1** Institut de Mathématiques de Bourgogne,
Université Bourgogne Franche-Comté, 21078 Dijon, France
**2** Faculty of Physics, University of Warsaw, ul. Pasteura 5, 02-093 Warsaw, Poland
**3** Interdisciplinary Center for Theoretical Study, University of Science and Technology of
China, Peng Huanwu Center for Fundamental Theory, Hefei, Anhui 230026, China
**4** Walter Burke Institute for Theoretical Physics,
California Institute of Technology, Pasadena, CA 91125, USA

## Abstract

We consider a large class of branes in toric strip geometries, both non-periodic and periodic ones. For a fixed background geometry we show that partition functions for such branes can be reinterpreted, on one hand, as quiver generating series, and on the other hand as wave-functions in various polarizations. We determine operations on quivers, as well as $SL(2,\mathbb{Z})$ transformations, which correspond to changing positions of these branes. Our results prove integrality of BPS multiplicities associated to this class of branes, reveal how they transform under changes of polarization, and imply all other properties of brane amplitudes that follow from the relation to quivers.



# 1    Introduction

Recently an intricate relation between open topological strings and quiver representation theory has been discovered. It states that to a given brane configuration in a toric Calabi-Yau threefold one can assign a quiver, whose various characteristics encode topological string data of the corresponding brane setup. For example, open topological string partition functions are identified with quiver generating series, BPS numbers are identified with motivic Donaldson-Thomas invariants (which proves integrality of the former invariants conjectured two decades ago), etc. To date, this correspondence has been analyzed in two specific contexts. Originally, its incarnation referred to as the knots-quivers correspondence was discovered in [1,2]. From topological string theory viewpoint, in this context one engineers a specific lagrangian brane in the resolved conifold, which represents some particular knot. Partition functions of such branes, or equivalently colored polynomials of a knot under consideration, can be written in the form of a quiver generating series, thereby making contact with quiver representation theory and leading to many nontrivial consequences. The knots-quivers correspondence was shown to hold for various knots up to 6 crossings and for some infinite series in [2,3], it was proven for two-bridge knots in [4] and for arborescent knots in [5]. Its relations to topological string theory were further discussed in [6–8], and related developments are presented in [9–12].

From the topological string perspective, the knots-quivers correspondence is concerned with presumably complicated lagrangian branes, in a relatively simple geometry of the resolved conifold. The second setup in which the above mentioned relation to quivers arises is opposite, and involves the basic Aganagic-Vafa branes embedded in more complicated toric Calabi-Yau threefolds. An interesting and quite tractable class of toric threefolds are those without compact four-cycles, referred to as strip geometries, for which the relation to quivers was revealed in [6]. In that work a specific lagrangian brane was considered, attached to one particular leg of a strip geometry.

In this paper we focus on the latter approach, i.e. lagrangian branes in strip geometries, and show that their quiver description is consistent with the interpretation of their partition functions as wave-functions in different polarizations. First, we analyze much more general brane configurations than in [6], i.e. we consider toric branes attached to any of the external legs of a strip geometry – including periodic toric geometries – and identify corresponding quivers. Second, for a pair of branes in a fixed underlying geometry, having identified their partition functions as a wave-function in different polarizations, we determine $SL(2,\mathbb{Z})$ transformations corresponding to a change of polarization (in other words a change of a position of a brane, i.e. a toric leg it is attached to).

Recall that an interpretation of the A-model open topological string partition function as a wave-function follows from a target space description in terms of Chern-Simons theory [13]. The phase space in this case is identified with the space of field configurations at the boundry of a lagrangian brane, in particular the boundary at infinity for non-compact branes, such as those considered in this paper. This viewpoint was elucidated in [14] – in particular, based on the observation for $\mathbb{C}^3$, it was postulated that not only brane partition functions transform as wave-functions, but also brane creation operators in the fermionic picture transform appropriately. This conjecture was verified in detail for branes in the conifold geometry in [15].

In this paper we generalize the results in [6] to arbitrary (non-periodic or periodic) strip geometries, and reinterpret them in terms of wave-function transformations in the spirit of [14, 15]. We determine quantum curves, also referred to as quantum A-polynomials [6, 16], as well as quivers, corresponding to geometries under consideration. While we consider much more general geometries than the conifold, the price we pay is that in most cases we identify an integral kernel that encodes canonical transformations between various partition functions based on the semi-classical analysis. We postulate that the same kernel encodes canonical transformations at the quantum level too. Furthermore, we reinterpret these transformations as relating various quiver descriptions of the corresponding branes. In particular, we find that changing a position of a brane in a toric geometry is represented in general only by the identity, $S^2$, or $T$ operation (or their composition), while a single $S$ operation may arise only for the special case of $\mathbb{C}^3$ or resolved conifold geometry.

We also stress that the identification of quivers corresponding to various branes that we consider implies that various properties discussed in [1, 2, 6] also hold. In particular, identification of quivers corresponding to toric branes of our interest proves that BPS multiplicities (i.e. LMOV invariants [17, 18]) associated to such branes are integer. Moreover, the algebra of such BPS states can be identified with the cohomological Hall algebra of the corresponding quiver. It would be interesting to analyze these properties in more detail, not only for strip geometries, but also for toric manifolds with compact four-cycles.

The plan of the paper is as follows. In section 2 we fix the notation, compute topological string amplitudes and corresponding quantum curves for branes in non-periodic and periodic toric strip geometries, and determine corresponding quivers. In section 3 we analyze transformations of brane partition functions under $SL(2,\mathbb{Z})$ transformations and their wave-function character. In section 4 we show how quiver A-polynomials transform under $SL(2,\mathbb{Z})$ transformations. In section 5 we illustrate these considerations in various examples, including both non-periodic and periodic strip geometries. In appendix A we provide more details on topological vertex formalism and compute amplitudes for branes attached to horizontal legs of a strip geometry.

## 2 Brane amplitudes and quivers

In this section we compute topological string amplitudes for branes in "strip" toric geometries, find corresponding quantum A-polynomials, and determine corresponding quivers.

### 2.1 Topological strings on the strip

In this paper we consider Calabi-Yau threefolds without compact four-cycles. They are also called "strip" geometries, since their toric diagrams arise from a triangulation of a strip [19]. An example of a dual web diagram is shown in fig. 1 (we will loosely call such web diagrams also as toric diagrams). Such diagrams can be thought of as a concatenation of trivalent vertices. Each finite segment in such a diagram (spanned between two neighboring vertices) is referred to as an internal leg and represents $\mathbb{P}^1$ with a Kähler parameter $Q_i$. In this work

we consider lagrangian (Aganagic-Vafa [18]) branes that are represented by thick segments attached to external legs of a toric diagram, see fig. 1.

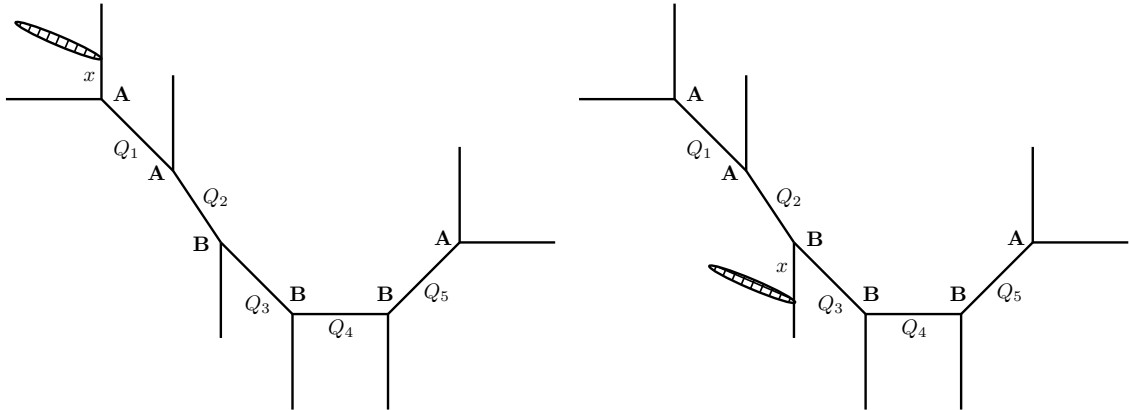

Figure 1: Examples of a strip geometry with branes on external legs. Kähler parameters are denoted by $Q_i$, each vertex is of type $A$ or $B$, and the brane modulus is denoted by $x$.

Closed and open topological string amplitudes for toric geometries can be calculated using the formalism of topological vertex [14,20]. For strip geometries this formalism can be simplified as discussed in [19], see also [6]. In appendix A we review this formalism and generalize it in order to take into account branes attached to non-vertical legs (which are attached to the first or the last vertex in a strip). Here let us simply recall, that in this simplified formalism we assign a type $A$ or $B$ to each vertex, in the following way. If a brane is attached to the first vertex, its type is chosen according to fig. 2; if a brane is attached neither to the first nor the last vertex, then the type of the first vertex is chosen as if a brane was attached to its vertical line, and a configuration with a brane attached to the last vertex is discussed in the appendix A.2. The types of the following vertices are assigned in such a way that the neighboring vertices have the same type if a segment connecting them represents local geometry of $\mathcal{O}(0) \oplus \mathcal{O}(-2)$, and they have opposite type if the local geometry is of $\mathcal{O}(-1) \oplus \mathcal{O}(-1)$ type. Apart from Kähler parameters $Q_i$, topological string amplitudes depend also on the coupling $\hbar$ which is captured by $q = e^{\hbar}$, and we assume that $|q| < 1$. Furthermore, open topological string amplitudes that involve a single brane depend an open modulus $x$ (or a number of such moduli, for more general brane configurations). Then, following the rules presented in appendix A, the open string partition function for a brane in framing $f$, attached to the $i$-th external vertical leg, takes form

$$\psi_{f,i} = \sum_{n=0}^{\infty} \left( (-1)^n q^{n(n-1)/2} \right)^{f+1} \frac{x^n}{(q;q)_n} \prod_{j<i} X_{ji} \prod_{j>i} X_{ij}, \tag{1}$$

where $X_{ij}$ are $q$-Pochhammer factors labeled by the indices of the distinguished vertex $i$ and of another vertex $j$, whose form is given in table 1. These factors depend on the types of vertices $i$ and $j$, and on the product of Kähler parameters $Q_{ij} = Q_i Q_{i+1} \cdots Q_{j-1}$. There are four types of these $q$-Pochhammer symbols: $(Q;q)_n$, $(Q;q)_n^{-1}$, $(Q;q^{-1})_n$, and $(Q;q^{-1})_n^{-1}$, whose numbers we denote respectively by $a$, $b$, $c$, and $d$. We also introduce the notation $r = a + c$ and $s = b + d$, and rewrite $q$-Pochhammers with the second argument $q^{-1}$ using the identity

$$(Q;q^{-1})_n = (-1)^n q^{-n(n-1)/2} Q^n (Q^{-1};q)_n. \tag{2}$$

With this notation, the expression (1) takes form

$$\psi_{f,i}(x) = \sum_{n=0}^{\infty} \left((-1)^n q^{n(n-1)/2}\right)^{f+1-c+d} \frac{\left(x \frac{\gamma_1 \cdots \gamma_c}{\delta_1 \cdots \delta_d}\right)^n}{(q;q)_n} \frac{(\alpha_1;q)_n \cdots (\alpha_a;q)_n (\gamma_1^{-1};q)_n \cdots (\gamma_c^{-1};q)_n}{(\beta_1;q)_n \cdots (\beta_b;q)_n (\delta_1^{-1};q)_n \cdots (\delta_d^{-1};q)_n},$$
(3)

where $\alpha_k, \beta_k, \gamma_k, \delta_k$ are given by a string $Q_{ij}$ of Kähler parameters and can be determined explicitly for a given strip geometry and for a given position of the brane. Note that there are $r$ $q$-Pochhammer symbols in the numerator and $s$ in the denominator. We also refer to $\psi_{f,i}(x)$ as the wave-function. For a special choice of framing $f = b - a$, the wave-function is simply a $q$-hypergeometric function

$$\psi_{b-a,i}(x) = {}_{a+c}\phi_{b+d} \begin{bmatrix} \alpha_1 & \cdots & \alpha_a & \gamma_1^{-1} & \cdots & \gamma_c^{-1} \\ \beta_1 & \cdots & \beta_b & \delta_1^{-1} & \cdots & \delta_d^{-1} \end{bmatrix}; q, x \frac{\gamma_1 \cdots \gamma_c}{\delta_1 \cdots \delta_d} \end{bmatrix},$$
(4)

which generalizes an analogous statement in [6].

We stress that expressions (1), (3), and (4) are relevant for branes attached to the $i$'th vertical leg. To obtain partition functions for branes attached to non-vertical legs (which arise for the first and the last vertex in the strip), one simply needs to consider the amplitude (1) for a brane on the same (the first or the last) vertex and change types of all vertices to the opposite ones. This essentially changes the form of $X_{ji}$ and $X_{ij}$ factors given in table 1. Effectively, if we start from the partition function for a brane on a vertical leg attached to the first or the last vertex, and replace $q$ by $q^{-1}$ in all factors $X_{ij}$ or $X_{ji}$, then we obtain the amplitude for a brane attached to the non-vertical leg.

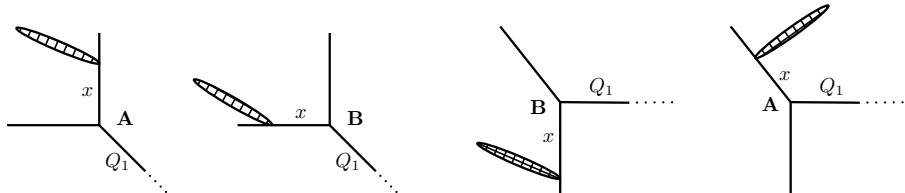

Figure 2: If a brane is attached to the first vertex, the type of this vertex depends on the external leg to which the brane is attached. If a brane is attached neither to the first or the last vertex, then the type of the first vertex is chosen as if the brane was on its vertical line (i.e. as in the first or the third picture). The case of a brane attached to the last vertex is discussed in the appendix.

Table 1: The contribution $X_{ij}$ to the open string partition function in a strip geometry with a single brane attached to an external leg of the $i$-th vertex. Vertices $i$ and $j$ are of type $A$ or $B$, and the underlined symbol denotes the position of the brane.

|   |   | $X_{ij}$, $i < j$ | $X_{ji}$, $i > j$ |
|---|---|---|---|
| $r$ | $a$ | $(\underline{A}, B) \rightarrow (Q_{ij};q)_n$ | $(A, \underline{B}) \rightarrow (Q_{ji};q)_n$ |
|   | $c$ | $(\underline{B}, A) \rightarrow (Q_{ij};q^{-1})_n$ | $(B, \underline{A}) \rightarrow (Q_{ji};q^{-1})_n$ |
| $s$ | $b$ | $(\underline{A}, A) \rightarrow (Q_{ij};q)_n^{-1}$ | $(B, \underline{B}) \rightarrow (Q_{ji};q)_n^{-1}$ |
|   | $d$ | $(\underline{B}, B) \rightarrow (Q_{ij};q^{-1})_n^{-1}$ | $(A, \underline{A}) \rightarrow (Q_{ji};q^{-1})_n^{-1}$ |

## 2.2 Quantum A-polynomials

Having determined the wave-function (3), it is not hard to derive a difference equation it satisfies. Writing $\psi_{f,i}(x) = \sum_n p_n x^n$, we find that the summand $p_n$ obeys

$$p_{n+1}\left(1-q^{n+1}\right)\prod_{i=1}^{b}(1-\beta_i q^n)\prod_{i=1}^{d}(\delta_i-q^n) = p_n(-1)^{f+1-c+d}q^{n(f+1+d-c)}\prod_{i=1}^{a}(1-\alpha_i q^n)\prod_{i=1}^{c}(\gamma_i-q^n). \tag{5}$$

By multiplying both sides by $x^{n+1}$, summing over all $n$, and introducing the operators $\hat{x}$ and $\hat{y}$ acting respectively by $\hat{x}f(x) = xf(x)$ and $\hat{y}f(x) = f(qx)$, we find the following operator

$$\widehat{A}(\hat{x},\hat{y}) = (1-\hat{y})\prod_{i=1}^{b}(1-q^{-1}\beta_i\hat{y})\prod_{i=1}^{d}(\delta_i-q^{-1}\hat{y})+(-1)^{f-c+d}\hat{x}\prod_{i=1}^{a}(1-\alpha_i\hat{y})\prod_{i=1}^{c}(\gamma_i-\hat{y})\hat{y}^{f+1+d-c}, \tag{6}$$

which we also refer to as a quantum curve or quantum A-polynomial [6,16], and which annihilates the wave-function

$$\widehat{A}(\hat{x},\hat{y})\psi_{f,i}(x) = 0. \tag{7}$$

In the classical limit $q \to 1$ the above operator reduces to the polynomial that defines the classical mirror curve $A(x,y) = 0$; it has genus zero and takes form

$$A(x,y) = (1-y)\prod_{i=1}^{b}(1-\beta_i y)\prod_{i=1}^{d}(\delta_i-y)+(-1)^{f-c+d}x\prod_{i=1}^{a}(1-\alpha_i y)\prod_{i=1}^{c}(\gamma_i-y)y^{f+1+d-c}. \tag{8}$$

## 2.3 Relation to quivers

In turn, we now introduce quivers and the associated motivic generating series. The structure of a quiver with $m$ vertices can be encoded in a matrix $C \in \mathbb{Z}^{m\times m}$, such that $C_{i,j}$ denotes the number of arrows from vertex $i$ to vertex $j$. A quiver is called symmetric if $C_{i,j} = C_{j,i}$. Various information about the moduli space of representations of a symmetric quiver is encoded in the corresponding motivic generating series that takes form [21–25]

$$P_C(x_1,\ldots,x_m) = \sum_{d_1,\ldots d_m}\frac{\left(-q^{1/2}\right)^{\sum_{i,j=1}^{m}C_{i,j}d_i d_j}}{(q;q)_{d_1}\cdots(q;q)_{d_m}}x_1^{d_1}\cdots x_m^{d_m}. \tag{9}$$

In particular, motivic Donaldson-Thomas invariants $\Omega_{d_1,\ldots d_m;j}$ that characterize such moduli spaces are determined by a decomposition of this generating series into quantum dilogarithms

$$P_C(x_1,\ldots,x_m) = \prod_{(d_1,\ldots,d_m)\neq 0}\prod_{j\in\mathbb{Z}}\prod_{k\geq 0}\left(1-\left(x_1^{d_1}\cdots x_m^{d_m}\right)q^{k+(j+1)/2}\right)^{(-1)^{j+1}\Omega_{d_1,\ldots,d_m;j}}. \tag{10}$$

In what follows we are often interested in the classical limit of the quiver generating series

$$P_C(x_1,\ldots,x_m) = \exp\left(\frac{1}{\hbar}S(x_1,\ldots,x_m) + \mathcal{O}(\hbar^0)\right). \tag{11}$$

To determine such a limit we introduce $z_i = e^{\hbar d_i} \in \mathbb{C}^*$, replace the sums over $d_i$ in (9) by integrals over $z_i$, rewrite the summand in the exponential form, and identify the action $S$ as the leading order term in the exponent

$$S = \frac{1}{2}\sum_{i,j=1}^{m}C_{i,j}\ln z_i \ln z_j + \sum_{j=1}^{m}\left(\ln z_j \ln(-1)^{C_{j,j}}x_j + \text{Li}_2(z_j) - \text{Li}_2(1)\right). \tag{12}$$

The variables $\{z_i\}_{i=1}^m$ are determined by the saddle point equations $\partial S/\partial z_i = 0$, which are analogous to the twisted F-term condition with the action $S$ playing a role of the twisted superpotential $\widehat{W}$. These equations take explicit form

$$1 = \exp\left(z_j \frac{\partial S}{\partial z_j}\right) = (-1)^{C_{j,j}} x_j \frac{\prod_{k=1}^m z_k^{C_{k,j}}}{1-z_j}. \tag{13}$$

The action $S$ can be also expressed purely through $z_j$

$$S = -\frac{1}{2} \sum_{i,j=1}^m C_{i,j} \ln z_i \ln z_j - \sum_{j=1}^m \mathrm{Li}_2(1-z_j), \tag{14}$$

where we used the Euler reflection formula

$$\mathrm{Li}_2(x) + \mathrm{Li}_2(1-x) = \mathrm{Li}_2(1) - \ln x \ln(1-x). \tag{15}$$

One of the main observations of this paper is the statement that the open string partition function (3) admits a representation in the form of the motivic generating series (9), for appropriate quiver matrix $C$ and appropriate identification of parameters. Such a quiver representation, for a brane attached to the first (left-most) vertex of a strip geometry, was found in [6]. We now find a generalization of that result to a brane attached to any leg in a toric strip diagram; namely, we find that the wave-function (3) can be written in the form

$$\psi_{f,i}(x) = P_C\left(x\frac{\gamma_1\cdots\gamma_c}{\delta_1\cdots\delta_d}, \boldsymbol{\alpha}_1,\ldots,\boldsymbol{\alpha}_a, \boldsymbol{\gamma}_1^{-1},\ldots,\boldsymbol{\gamma}_c^{-1}, \boldsymbol{\beta}_1,\ldots,\boldsymbol{\beta}_b, \boldsymbol{\delta}_1^{-1},\ldots,\boldsymbol{\delta}_d^{-1}\right), \tag{16}$$

where we use the following notation for the arguments of the motivic generating series

$$\boldsymbol{\alpha} = (q^{-1/2}\alpha, \alpha), \quad \boldsymbol{\alpha}^{-1} = (q^{-1/2}\alpha^{-1}, \alpha^{-1}), \tag{17}$$

and similarly for $\boldsymbol{\beta}, \boldsymbol{\gamma}, \boldsymbol{\delta}$. The quiver that determines (16) is represented by the following matrix of size $m = 1 + 2r + 2s$

$$C = \left[\begin{array}{c|c|c} f+1-c+d & \boldsymbol{v}_1^{2r} & \boldsymbol{v}_2^{2s} \\ \hline (\boldsymbol{v}_1^{2r})^T & A^{2r} & O^{2r,2s} \\ \hline (\boldsymbol{v}_2^{2s})^T & O^{2s,2r} & A^{2s} \end{array}\right], \tag{18}$$

where

$$\boldsymbol{v}_1^{2n} = \underbrace{(0,1,0,1,...,0,1)}_{2n}, \quad \boldsymbol{v}_2^{2n} = \underbrace{(1,0,1,0,...,1,0)}_{2n}, \quad A^{2n} = \mathrm{diag}[\underbrace{1,0,1,0,...,1,0}_{2n}], \tag{19}$$

and $O^{2m,2n}$ is the $2m \times 2n$ matrix with all elements being zero, while $T$ in the superscript denotes the transposition of a vector.

The above quiver generating series in fact factorizes into a number of universal $q$-Pochhammer factors and the remaining part, which can be written in terms of a smaller quiver matrix of size $1 + r + s$

$$\psi_{f,i}(x) = \frac{\prod_{j=1}^a (\alpha_j; q)_\infty \prod_{j=1}^c (\gamma_j^{-1}; q)_\infty}{\prod_{j=1}^b (\beta_j; q)_\infty \prod_{j=1}^d (\delta_j^{-1}; q)_\infty} \times$$
$$\times P_{C'}\left(x\frac{\gamma_1\cdots\gamma_c}{\delta_1\cdots\delta_d}, \alpha_1,\ldots,\alpha_a, \gamma_1^{-1},\ldots,\gamma_c^{-1}, q^{1/2}\beta_1,\ldots,q^{1/2}\beta_b, q^{1/2}\delta_1^{-1},\ldots,q^{1/2}\delta_d^{-1}\right), \tag{20}$$

where

$$C' = \left[ \begin{array}{c|c|c} f + 1 - c + d & \boldsymbol{v'}^r & \boldsymbol{v'}^s \\ \hline (\boldsymbol{v'}^r)^T & \boldsymbol{O}^{r,r} & \boldsymbol{O}^{r,s} \\ \hline (\boldsymbol{v'}^s)^T & \boldsymbol{O}^{s,r} & \boldsymbol{A'}^s \end{array} \right], \tag{21}$$

and

$$\boldsymbol{v'}^n = \underbrace{(1, 1, ..., 1)}_{n}, \qquad \boldsymbol{A'}^n = \text{diag}[\underbrace{1, 1, ..., 1}_{n}]. \tag{22}$$

Note that the size of a quiver represented by $C$ or $C'$ depends only on the number of vertices in the strip geometry, and not on the location of a brane. However, the structure of the quiver generating function depends on the number of $q$-Pochhammer factors in the numerator and denominator (respectively $r$ and $s$), which do depend on the brane location. The rules of table 1 imply that if we change the brane position from one vertex to another of the same type, the quiver stays intact. It changes, if we change the brane position to a vertex of another type.

In fact, the quiver matrix (18) is the same as the one found in [6] for a brane attached to the first vertex. Therefore more general brane partition functions that we find now (16) differ from those in [6] just by rescalings of quiver generating parameters. In particular this implies, that open BPS invariants (LMOV invariants) assigned to the branes that we consider now are given by an analogous formula as in [6], up to some straightforward redefinitions.

Let us also note how the action (12) simplifies for quivers corresponding to strip geometries (18). In this case quiver matrices have non-zero elements only in the first row and column and on the diagonal, so we can write

$$S = -\frac{1}{2}C_{1,1}(\ln y)^2 - \text{Li}_2(1 - y) + \sum_{j=2}^{m} S_j, \tag{23}$$

where

$$S_j = -C_{1,j} \ln y \ln z_j - \frac{C_{j,j}}{2}(\ln z_j)^2 - \text{Li}_2(1 - z_j), \qquad j = 2, \dots, m \tag{24}$$

and

$$y = \exp\left(x \frac{\partial S}{\partial x}\right). \tag{25}$$

Furthermore, looking in detail at non-zero entries, we find that

$$S = \frac{c - d - f - 1}{2}(\ln y)^2 - \text{Li}_2(1 - y) + \sum_{j=1}^{a} s(\alpha_j) + \sum_{j=1}^{c} s(\gamma_j^{-1}) - \sum_{j=1}^{b} s(\beta_j) - \sum_{j=1}^{d} s(\delta_j^{-1}), \tag{26}$$

where

$$s(x_j) = -\ln y \ln(1 - x_j y) - \text{Li}_2(x_j y) + \text{Li}_2(x_j). \tag{27}$$

From this action we can also compute a general form of the A-polynomial

$$(1 - y) \prod_{j=1}^{b} (1 - \beta_j y) \prod_{j=1}^{d} (1 - \delta_j^{-1} y) = (-1)^{C_{1,1}} x y^{C_{1,1}} \times \prod_{j=1}^{a} (1 - \alpha_j y) \prod_{j=1}^{c} (1 - \gamma_j^{-1} y), \tag{28}$$

which is consistent with (8), upon a correct specialization of variables.

## 2.4 Periodic chain geometry

One important class of examples that we consider in this work are toric manifolds encoded in periodic toric diagrams. Among others, using such manifolds one can engineer various interesting supersymmetric gauge theories [26]. In what follows we consider wave-function transformations for periodic chain geometries defined by identifying the external line along the horizontal axis, as in fig. 3. It is sufficient to focus on geometries with arbitrary length $2N$ (i.e. with $2N$ Kähler parameters $Q_i, \tilde{Q}_i$ for $i = 1, \ldots, N$), with an interlacing pattern of vertices $ABAB \ldots$. Indeed, note that all compactified geometries are related to those with vertices of type $ABAB \ldots$ by flop transitions, for the following reason. In order to glue the external legs along the horizontal line, the corresponding Newton polygon must be a parallelogram, and in this case we can compactify the horizontal line by identifying the upper and bottom corners of the parallelogram. By utilizing the periodicity, geometries encoded in various parallelograms are all equivalent to the geometry encoded in a rectangle, and its all triangulations can be related by flop transitions to the triangulation which corresponds to $ABAB \ldots$ geometry. Therefore understanding the class of $ABAB \ldots$ geometries is sufficient.

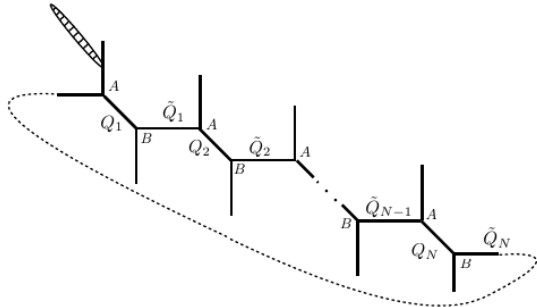

Figure 3: The periodic chain geometry with a lagrangian brane. The dashed line denotes the compactification along the horizontal line.

Consider the open string partition function for a brane attached to the first vertex. It has also been calculated in [27], and takes form

$$\psi_f(x) = \sum_{n=0}^{\infty} \left((-1)^n q^{n(n-1)/2}\right)^{f+1} \frac{x^n}{(q;q)_n} \prod_{m=1}^{\infty} \prod_{b=1}^{N} \frac{(P_b p^{m-1};q)_n (P_b^{-1} p^m; q^{-1})_n}{(\tilde{P}_b p^{m-1};q)_n (\tilde{P}_b' p^m; q^{-1})_n}, \tag{29}$$

where $p = \prod_{i=1}^{N} Q_i \tilde{Q}_i$, and using Kronecker $\delta_{i,j}$ we have

$$P_b = Q_b \prod_{k=1}^{b-1} Q_k \tilde{Q}_k, \quad \tilde{P}_b = (qp)^{\delta_{1,b}} \prod_{k=1}^{b-1} Q_k \tilde{Q}_k, \quad \tilde{P}_b' = q^{-\delta_{1,b}} \prod_{k=1}^{b-1} (Q_k \tilde{Q}_k)^{-1}. \tag{30}$$

We find that this wave-function is annihilated by the following quantum mirror curve

$$\widehat{H}(\widehat{x}, \widehat{y}) = \prod_{b=1}^{N} \theta\left(q^{-\delta_{1,b} + \sum_{j=2}^{N} \delta_{j,b}} \tilde{P}_b \widehat{y}\right) + (-1)^f \widehat{x} \widehat{y}^{f+1} \prod_{b=1}^{N} \theta\left(P_b \widehat{y}\right), \tag{31}$$

where we define the theta function

$$\theta(x) = (x;p)_{\infty} (px^{-1};p)_{\infty}. \tag{32}$$

Note that in the limit $p \to 0$ we have $\theta(x) \to (1-x)$, i.e. a periodic chain becomes a non-periodic one.

Furthermore, we find that (29) can be written in a form of a quiver generating function, with a quiver of infinite size. To this end we use first the $\zeta$-function regularization and rewrite (29) in the form

$$\psi_f(x) = \sum_{n=0}^{\infty} \left((-1)^n q^{n(n-1)/2}\right)^{f+1} \frac{\left(\prod_{b=1}^{N} (P_b \tilde{P}'_b)^{1/2} x\right)^n}{(q;q)_n} \prod_{m=1}^{\infty} \prod_{b=1}^{N} \frac{(P_b p^{m-1};q)_n (P_b p^{-m};q)_n}{(\tilde{P}_b p^{m-1};q)_n (\tilde{P}'^{-1}_b p^{-m};q)_n}$$

$$= P_C \Big( \prod_{b=1}^{N} (P_b \tilde{P}'_b)^{1/2} x, \boldsymbol{P}, \boldsymbol{R}, \boldsymbol{S}, \boldsymbol{T} \Big), \tag{33}$$

where

$$\boldsymbol{P} = \left\{ q^{-1/2} P_b p^{m-1}, P_b p^{m-1}, b = 1, ..., N, \ m = 1, ..., \infty \right\}, \tag{34a}$$

$$\boldsymbol{R} = \left\{ q^{-1/2} P_b p^{-m}, P_b p^{-m}, b = 1, ..., N, \ m = 1, ..., \infty \right\}, \tag{34b}$$

$$\boldsymbol{S} = \left\{ q^{-1/2} \tilde{P}_b p^{m-1}, \tilde{P}_b p^{m-1}, b = 1, ..., N, \ m = 1, ..., \infty \right\}, \tag{34c}$$

$$\boldsymbol{T} = \left\{ q^{-1/2} \tilde{P}'^{-1}_b p^{-m}, \tilde{P}'^{-1}_b p^{-m}, b = 1, ..., N, \ m = 1, ..., \infty \right\}, \tag{34d}$$

and the quiver matrix $C$ is given by an infinitely large generalization of (18)

$$C = \begin{bmatrix} \begin{array}{c|c|c} f+1 & \boldsymbol{v}_1^n & \boldsymbol{v}_2^n \\ \hline (\boldsymbol{v}_1^n)^T & \boldsymbol{A}^n & \boldsymbol{O}^{n,n} \\ \hline (\boldsymbol{v}_2^n)^T & \boldsymbol{O}^{n,n} & \boldsymbol{A}^n \end{array} \end{bmatrix}_{n \to \infty}. \tag{35}$$

Note that infinite products over $m = 1, \ldots, \infty$ in (33) can be written in the form of the theta function, and then the wave-function can be expressed as an elliptic hypergeometric function. The regularized wave-function (33) is annihilated by the quantum A-polynomial

$$\widehat{A}(\hat{x}, \hat{y}) = (1 - \hat{y}) \prod_{b=1}^{N} (q \tilde{P}_b \hat{y}; p)_\infty (q \tilde{P}'^{-1}_b p^{-1} \hat{y}; p^{-1})_\infty +$$

$$+ (-1)^f \prod_{b=1}^{N} (P_b \tilde{P}'_b)^{1/2} \hat{x} \hat{y}^{f+1} \prod_{b=1}^{N} (P_b \hat{y}; p)_\infty (P_b p^{-1} \hat{y}; p^{-1})_\infty, \tag{36}$$

and this operator is a regularized form of (31).

# 3 Wave-function behavior and $SL(2,\mathbb{Z})$ transformations

In [14, 15] it was conjectured that topological string partition functions for branes in toric Calabi-Yau manifolds have wave-function character. This implies that partition functions for different branes in a fixed toric geometry can be regarded as wave-functions in different polarizations, and can be related by $SL(2,\mathbb{Z})$ transformations. Such a transformation, corresponding to a change of a position of a brane from the $i$'th to the $j$'th vertex, can be implemented via an integral transform

$$\psi_j(x_j) \sim \int_{\mathcal{C}_{ij}} dx_i K(x_j, x_i) \psi_i(x_i), \tag{37}$$

where $\mathcal{C}_{ij}$ is an appropriate contour and the kernel

$$K(x_i, x_j) = \exp\left( \frac{1}{2c\hbar} \left( d(\ln x_i)^2 - 2\ln x_i \ln x_j + a(\ln x_j)^2 \right) \right) \tag{38}$$

is determined by $a, c, d \in \mathbb{Z}$ that specify an element of $SL(2, \mathbb{Z})$

$$K = \begin{pmatrix} a & (ad-1)/c \\ c & d \end{pmatrix}. \tag{39}$$

Recall that all elements of $SL(2, \mathbb{Z})$ are generated by two generators

$$S = \begin{pmatrix} 0 & 1 \\ -1 & 0 \end{pmatrix} \qquad T = \begin{pmatrix} 1 & 0 \\ 1 & 1 \end{pmatrix}, \tag{40}$$

which satisfy $S^2 = -1$ and $(ST)^3 = -1$.

In [15] the above statement was verified for $\mathbb{C}^3$ and the conifold geometry. To this end an appropriate framing was chosen, for which open topological string partition functions for these two manifolds are captured by a finite number of LMOV invariants and can be written as a product of a finite number of quantum dilogarithms. In this case the integral in (37) can be evaluated, at least formally, in the exact form, by the analysis of the $q$-expansion of open partition functions.

In this work we are going to confirm the above conjecture for arbitrary strip geometries, including periodic chains, for branes at various positions and in arbitrary framing, and also to provide quiver interpretation of these transformations. For arbitrary strip geometries the number of LMOV invariants is infinite, and the integral (37) cannot be evaluated exactly. However, we will verify the transformation rule (37) in the classical limit, which is possible in such a general case. The analysis in this section relies on the quiver representation of brane partition functions discussed in section 2.3 – namely, we identify which $SL(2, \mathbb{Z})$ operations preserve such an underlying quiver structure of the brane wave-function.

In the quiver interpretation (16), the open modulus $x$ of a brane is identified with the quiver generating parameter $x_1$. Let us therefore consider the wave-function identified as

$$\psi(x) = P_C(x, x_2, \ldots, x_m) = \exp\left(\frac{1}{\hbar}S(x) + \mathcal{O}(\hbar^0)\right), \tag{41}$$

where the action $S(x)$ is the corresponding specialization of (12). A function that is a solution to the classical A-polynomial equation (mirror curve equation) $A(x, y) = 0$ takes form

$$y(x) = \lim_{\hbar \to 0} \frac{\psi(qx)}{\psi(x)} = \lim_{\hbar \to 0} \frac{P_C(qx, x_2, \ldots, x_m)}{P_C(x, x_2, \ldots, x_m)} = \exp\left(x\frac{\partial S}{\partial x}\right). \tag{42}$$

For $\psi(x)$ in (41) and a general kernel (39), the transformation (37) in the classical limit takes form (for a similar analysis see e.g. [28])

$$\int dx K(x', x)\psi(x) = \int dx \exp\left(\frac{1}{\hbar}\left(S(x) + \frac{1}{2c}\left(d(\ln x)^2 - 2\ln x \ln x' + a(\ln x')^2\right)\right)\right). \tag{43}$$

The saddle point condition reads

$$\frac{\partial S(x)}{\partial x} + \frac{d}{c}\frac{\ln x}{x} - \frac{1}{c}\frac{\ln x'}{x} = 0. \tag{44}$$

From the relation (42) we get $x' = x^d y^c$ (for $x \neq 0$). Solving this relation for $x$ yields $x = x(x')$. The result of the integration is then

$$\int dx K(x', x)\psi(x) = \exp\left(\frac{1}{\hbar}S'(x')\right), \tag{45}$$

where we define the transformed action $S'(x')$

$$S'(x') = S(x(x')) + \frac{1}{2c}\left(d(\ln x(x'))^2 - 2\ln x(x')\ln x' + a(\ln x')^2\right). \tag{46}$$

It then follows that

$$y' = \exp\left(x'\frac{\partial S'(x')}{\partial x'}\right) = x^{-1/c}x'^{a/c} = x^{(da-1)/c}y^a. \tag{47}$$

Altogether we get

$$\begin{pmatrix} y' \\ x' \end{pmatrix} = \begin{pmatrix} y^a x^{(ad-1)/c} \\ y^c x^d \end{pmatrix} \equiv K \star \begin{pmatrix} y \\ x \end{pmatrix}, \tag{48}$$

where $K \in SL_2(2,\mathbb{Z})$ is given in (39) and we introduced the notation

$$\begin{pmatrix} \alpha & \beta \\ \gamma & \delta \end{pmatrix} \star \begin{pmatrix} y \\ x \end{pmatrix} = \begin{pmatrix} y^\alpha x^\beta \\ y^\gamma x^\delta \end{pmatrix}. \tag{49}$$

The above analysis shows how a wave-function $\psi(x)$ and a pair $(x,y)$ are transformed into $\psi'(x')$ and $(x',y')$ on the classical level. However, for a given kernel $K$, there is no guarantee that the resulting wave-function $\psi'(x')$ can be also written in the quiver form. Nonetheless, we know that brane partition functions can always be written in quiver form (16). It follows that not all, but only some special subset of $SL(2,\mathbb{Z})$ transformations, corresponds to changing a brane location. Let us now show that transformations that lead to $\psi'(x')$, and which can be written in a quiver form, are combinations of $T$ and $S^2$ operations, while transformations generated by $S$ arise only in a very special case.

Consider first $T$ transformation. Applying it $n$ times yields $\begin{pmatrix} y' \\ x' \end{pmatrix} = T^n \star \begin{pmatrix} y \\ x \end{pmatrix} = \begin{pmatrix} y \\ xy^n \end{pmatrix}$, which corresponds to a change of framing by $n$. This transformation changes only $x$ variables and modifies only the first saddle-point equation, which then turns into

$$1 - y' = (-1)^{C_{1,1}} x'(y')^{C_{1,1}-n} \prod_{j=2}^{m} z_j^{C_{j,k}}. \tag{50}$$

This transformed equation corresponds to a quiver $C'$ such that $C'_{j,k} = C_{j,k} - n\delta_{j,1}\delta_{j,k}$ (and with an additional rescaling of the $x'_1$ by $(-1)^f$). For us it is crucial that the transformed quiver $C'$ exists, which asserts that $T^n$ is an allowed transformation. Equivalently, this can be argued by the analysis of how the classical action transforms. The original action (23) after $T^n$ transformation takes form

$$S'(x') = S(x(x')) + \frac{1}{2n}(\ln x(x')/x')^2 = -\frac{1}{2}(C_{1,1}-n)(\ln y')^2 - \text{Li}(1-y') +$$
$$-\sum_{i=2}^{m} C_{1,i}\ln y'\ln z_i - \frac{1}{2}\sum_{i,j=2}^{m} C_{i,j}\ln z_i\ln z_j - \sum_{j=2}^{m} \text{Li}_2(1-z_j), \tag{51}$$

which indeed corresponds to the action associated to the quiver $C'$ given above. For the motivic generating series we therefore confirm (at least to the leading order) the relation

$$(T^n\psi_C)(x) = \psi_{C'}((-1)^n x). \tag{52}$$

Consider now a single $S$ transformation $\begin{pmatrix} y' \\ x' \end{pmatrix} = S \star \begin{pmatrix} y \\ x \end{pmatrix} = \begin{pmatrix} x \\ y^{-1} \end{pmatrix}$. The transformed action takes form $S'(x') = S(x(x')) + \ln x \ln x'$ and exchanges the role of $x$ and $y$, so that it is difficult to infer about the general structure of $S'(x')$. To get a better understanding of the problem let

us consider a few examples. First, consider an A-polynomial which is symmetric in $x$ and $y$. Inspecting the general form (8) of the A-polynomial for the strip geometries we see that such a symmetry requires that the kernel (39) and framing $f$ are of the form $b = d = 0$, $f = c-1$ and either $a = 1, c = 0$ or $a = 0, c = 1$. In these two cases, A-polynomials read respectively

$$A(x,y) = 1 - y - x(1 - \alpha y), \qquad A(x,y) = 1 - y - x(\gamma - y). \tag{53}$$

These A-polynomials correspond to the conifold, which we analyze in more detail in section 5.2. Therefore we see that $S$ transformation is relevant at least for the conifold geometry.

In turn, consider two simple examples associated to a quiver with one node. For $C = (0)$ the saddle-point equation takes form $1-y = x$ and the classical action reads $S(x) = -\text{Li}_2(1-y) = -\text{Li}_2(x)$. After $S$ transformation they turn into $1 - y' = \frac{1}{x'}$ and

$$S'(x') = S + \ln x \ln x' = -\text{Li}_2(1 - 1/x') - \ln(1 - 1/x')\ln(1/x') = \text{Li}_2(1/x') - \text{Li}_2(1). \tag{54}$$

This action is related to a one-vertex quiver $C' = (1)$. Indeed, for this quiver the saddle-point equation takes form $y = 1/(1-x)$, and the classical action reads $S(x) = -\frac{1}{2}(\ln y)^2 - \text{Li}_2(1-y) = \text{Li}_2(1 - 1/y) = \text{Li}_2(x)$. Ultimately, $S$ transformation relates quiver (0) to quiver (1) and inverts the argument

$$(S\psi_{(0)})(x) = e^{\text{Li}_2(1)/\hbar}\psi_{(1)}^{-1}(x^{-1}). \tag{55}$$

As the second example of a quiver with one vertex, consider the case $C = (2)$. The saddle point equation reads $1 - y = xy^2$ and the action takes form $S(x) = -(\ln y)^2 - \text{Li}_2(1-y)$. After $S$ transformation we get the corresponding saddle-point equation $y' = x'(x'-1)$ and action $S'(x) = -(\ln x)^2 - \text{Li}_2(1-x)$. In this case the resulting curve $y' = y'(x')$ does not satisfy the condition $y'(0) = 1$, which quiver A-polynomials should necessarily satisfy. This means that in this case the $S$ transformation yields a wave-function which is not associated to a quiver. This shows that $S$ operation corresponds to changing a brane position only in some special cases (like the conifold geometry mentioned above), but not in general.

Finally, consider $S^2$ transformation $\binom{y'}{x'} = S^2 \star \binom{y}{x} = \binom{y^{-1}}{x^{-1}}$. It corresponds to the element (39) with $a = d = -1$ and then setting $c = 0$, so that $(ad-1)/c = 0$. The corresponding kernel can be identified as $K = \exp\left(-\frac{1}{2c\hbar}(\ln(xx'))^2\right)$, where we keep $c \neq 0$ as a regulator. In the limit $c \to 0$ the kernel localizes the integration to $\ln(xx') = 1$, which indeed yields $x' = x^{-1}$. The result of the action of $S^2$ is then

$$\psi'(x') = (S^2\psi)(x') \sim \exp\left(\frac{1}{\hbar}S(x'^{-1})\right). \tag{56}$$

Therefore, the resulting partition functions is described by the same quiver, merely evaluated at the reciprocal point. The transformed action can be rewritten as

$$S'(x') = S(1/x') =$$
$$= -\frac{1}{2}C_{1,1}(\ln y)^2 - \text{Li}(1 - 1/y) + \sum_{i=2}^{m}C_{1,i}\ln y \ln z_i - \frac{1}{2}\sum_{i,j=2}^{m}C_{i,j}\ln z_i \ln z_j - \sum_{j=2}^{m}\text{Li}_2(1 - z_j)$$
$$= -\frac{1}{2}(C_{1,1} + 1)(\ln y)^2 + \text{Li}(1 - y) + \sum_{i=2}^{m}C_{1,i}\ln y \ln z_i - \frac{1}{2}\sum_{i,j=2}^{m}C_{i,j}\ln z_i \ln z_j - \sum_{j=2}^{m}\text{Li}_2(1 - z_j),$$
$$\tag{57}$$

where we used that $y(x'^{-1}) = y'(x')^{-1}$ and the Landen's identity

$$\text{Li}(1 - z) + \text{Li}(1 - 1/z) = -\frac{1}{2}(\ln z)^2, \qquad z > 0. \tag{58}$$

This transformed action is encoded by the same quiver, simply with the argument evaluated at the reciprocal point. We can also check that the resulting form of the partition function is consistent with $y' = y^{-1}$. We have $y' = \lim_{q \to 1} \frac{\psi'(qx)}{\psi'(x)} = \exp(x' \partial_{x'} S(x'^{-1})) = \exp(-x \partial_x S(x)) = y^{-1}$, as expected.

Finally, let us also note that to move a brane around a strip geometry we need to slightly enlarge a set of allowed transformations, by including rescaling of the variables Kähler parameters. A general transformation that we need to consider takes form

$$y' = Q y^a x^{(ad-1)/c}, \qquad x' = \overline{Q} y^c x^d. \tag{59}$$

It is not hard to see that corresponding integration kernel, generalizing (38), takes form

$$K(x, x') = \exp\left( \frac{1}{2c\hbar}(d(\ln x)^2 - 2\ln x \ln(\overline{Q}^{-1}x') + a(\ln(\overline{Q}^{-1}x'))^2 + \frac{1}{\hbar}\log Q \log x' \right). \tag{60}$$

The saddle-point equation with this new kernel reads $cx\partial_x S + d\ln x - \frac{1}{c} - \ln(\overline{Q}^{-1}x') = 0$, which indeed leads to the correct relation for $x'$ in (59). The resulting classical action is

$$S'(x') = S(x(x')) + \frac{1}{2c}\left( d(\ln x(x'))^2 - 2\ln x(x')\ln(\overline{Q}^{-1}x') + a(\ln(\overline{Q}^{-1}x'))^2 \right) + \log Q \log x'. \tag{61}$$

From the transformed action we find, also in agreement with (59)

$$y' = \exp\left( x'\frac{\partial S'(x')}{\partial x'} \right) = \exp\left( -\frac{1}{c}\ln x + \frac{a}{c}\ln(\overline{Q}^{-1}x') + \log Q \right) = Q y^a x^{(ad-1)/c}. \tag{62}$$

In conclusion, only a subgroup of transformations generated by $T$ and $S^2$ acting on a quiver generating function results also in a quiver generating function, associated to a transformed quiver, possibly with rescaled variables. This conclusion is consistent with the behavior of quantum A-polynomials analyzed in the next section.

# 4 (Quantum) A-polynomials and $SL(2, \mathbb{Z})$ transformations

In this section we confirm the wave-function character of brane partition functions from another viewpoint, by considering the effect of $SL(2, \mathbb{Z})$ transformations on quantum A-polynomials. We first argue that, in general, only $T$ and $S^2$ transformations preserve the structure of A-polynomials and thus are allowed. Subsequently, we explicitly identify the operations that correspond to changing the position of a brane. We show that moving a brane from one toric leg into another one can be interpreted as an action of the following elements of $SL(2, \mathbb{Z})$: the identity when a brane is moving between two vertices of the same type, $S^2$ when it is moving between vertices of different types, and $T^{s-r}$ when it is moving between two external legs of the first or the last vertex in a strip. Note that in general there is no operation on a brane that corresponds to the action of a single generator $S$; as we show below, such an operation arises exceptionally only for the $\mathbb{C}^3$ or conifold geometry. These results agree with what we found in section 3.

## 4.1 Action of $SL(2, \mathbb{Z})$ on A-polynomials

We consider first how mirror curves, or equivalently quiver A-polynomials, derived in section 2.2, transform under the action of $SL(2, \mathbb{Z})$. We consider the action of generators $S$ and $T$ and the action of $S^2$. Under the action of $T$, the operators $(\hat{x}, \hat{y})$ transform as

$$(\hat{x}, \hat{y}) \xrightarrow{T} (\hat{x}\hat{y}, \hat{y}), \tag{63}$$

which corresponds to increasing framing by one, which can be always conducted.

The action of $S$, which takes form

$$(\widehat{x}, \widehat{y}) \xrightarrow{S} (\widehat{y}^{-1}, \widehat{x}), \tag{64}$$

is more subtle. It is sufficient to consider the classical case. Note that A-polynomials found in section 2.2 have the following general form

$$A(x, y) = P(y) + xQ(y), \tag{65}$$

so that after the $S$-transformation we get

$$A(x, y) \xrightarrow{S} P(x) + y^{-1}Q(x) = y^{-1}\big(Q(x) + yP(x)\big). \tag{66}$$

For the right hand side to be also of the form (65) we need to ensure that

$$Q(x) = C_1 W(x)(1 + \epsilon_1 x), \qquad P(x) = C_2 W(x)(1 + \epsilon_2 x), \tag{67}$$

where $C_i$ and $\epsilon_i$ are some constants and $W(x)$ is another polynomial in $x$, so that the resulting polynomial (up to an overall factor) is of order $x$

$$y^{-1}(Q(x) + yP(x)) = \frac{C_1 W(x)}{y}\Big(1 + \frac{C_2}{C_1}y + C_1\epsilon_1 x(1 + \frac{C_2\epsilon_2}{C_1\epsilon_1}y)\Big). \tag{68}$$

The question is then, when the condition (67) can be fulfilled. We have

$$\frac{Q(x)}{P(x)} = \frac{C_1}{C_2}\frac{1 + \epsilon_1 x}{1 + \epsilon_2 x} = (-1)^{f-c+d}\frac{x^{f+1+d-c}}{1-x}\frac{\prod_{i=1}^{a}(1 - \alpha_i x)\prod_{i=1}^{c}(\gamma_i - x)}{\prod_{i=1}^{b}(1 - \beta_i x)\prod_{i=1}^{d}(\delta_i - x)}. \tag{69}$$

First, we need to fix the framing as $f = c - d - 1$, so that the leading term is 1. Then, for the equality to hold, the necessary condition is that the orders of the rational functions match. For $\epsilon_i \neq 0$ this leads to the condition $a + c - b - d = 1$. Additionally, for $b, d \neq 0$ this requires pairwise cancellations and therefore cannot hold for arbitrary values of Kähler parameters. Analogous arguments exclude special cases $\epsilon_1 = 0$ or $\epsilon_2 = 0$.

Finally, consider the case $\epsilon_i \neq 0$ with $b = d = 0$. In that case, given that $a + c - b - d = 1$, we have two possibilities, either $(a = 1, c = 0)$ or $(a = 0, c = 1)$, and we find respectively

$$\frac{C_1}{C_2}\frac{1 + \epsilon_1 x}{1 + \epsilon_2 x} = -\frac{1 - \alpha x}{1 - x}, \qquad \frac{C_1}{C_2}\frac{1 + \epsilon_1 x}{1 + \epsilon_2 x} = -\frac{\gamma - x}{1 - x}. \tag{70}$$

These conditions are solved by $(\frac{C_1}{C_2} = 1, \epsilon_1 = -\alpha, \epsilon_2 = -1)$ or respectively $(\frac{C_1}{C_2} = -\gamma^{-1}, \epsilon_1 = -\gamma^{-1}, \epsilon_2 = -1)$. The $S$ transformation then yields

$$(a = 1, c = 0): \quad 1 - y - x(1 - \alpha y) \xrightarrow{S} y^{-1}(1 - y - x(\alpha - y)), \tag{71}$$

$$(a = 0, c = 1): \quad 1 - y - x(\gamma - y) \xrightarrow{S} -\gamma y^{-1}\big((1 - \gamma^{-1}y) - \gamma^{-1}x(1 - y)\big), \tag{72}$$

and in the second case further rescaling $x \to \gamma x$ and $y \to \gamma y$ brings the A-polynomial to the canonical form $-y^{-1}((1 - y) - x(1 - \gamma y))$. Note that these two special cases correspond simply to the conifold geometry, which was the main example analyzed in [15]; for more complicated strip geometries branes do not transform simply under the $S$ operation. Furthermore, note that for these two special cases, $S^2$ has the same effect as the identity transformation and produces the original A-polynomial.

Let us also consider the action of $S^2$ for a generic strip geometry

$$(\widehat{x}, \widehat{y}) \xrightarrow{S^2} (\widehat{x}^{-1}, \widehat{y}^{-1}), \tag{73}$$

for which the quantum A-polynomial (6) transforms into

$$\widehat{x}^{-1}\left[\prod_{i=1}^{c}(1-\gamma_i\widehat{y})\prod_{i=1}^{a}(\alpha_i-\widehat{y})+(-1)^{f-1-b-a}\widehat{x}(1-\widehat{y})\prod_{i=1}^{d}(1-\delta_i q\widehat{y})\prod_{i=1}^{b}(q^{-1}\beta_i-\widehat{y})\right]\widehat{y}^{-f-1-d-a}. \tag{74}$$

This is almost the correct form, apart from the absence of $(1-\widehat{y})$ in the first term and the presence of $(1-\widehat{y})$ in the second term. This can be fixed by rescaling $\widehat{y}$ by $\gamma_i^{-1}$ or $\alpha_i$ and extracting $(1-\widehat{y})$ in front of the product. At the same time the other "wrong" factor can be incorporated in one of the products, bringing the expression to the right form. Therefore $S^2$ transforms a geometry characterized by $(r,s)$ into a geometry characterized by $(r'=s+1, s'=r-1)$.

To sum up, the structure of A-polynomials is always preserved by $T$ and $S^2$ transformations (and their compositions). This structure in general is not preserved by a single $S$ operation, which makes sense only for $\mathbb{C}^3$ or resolved conifold geometry.

## 4.2 Moving to another vertex

We analyze now how moving a brane from one vertex to another is represented by elements of $SL(2,\mathbb{Z})$. Denote by $\widehat{A}_i(\widehat{x}_i, \widehat{y}_i)$ the quantum A-polynomial annihilating the wave function with the brane at the external leg of the $i$-th vertex. First, consider moving a brane one vertex ahead, from the $i$'th vertex to the $(i+1)$'th; these vertices can be of type $A$ or $B$.

Consider the case when both vertices are of type $A$. We distinguish contributions from vertices preceding the vertex $i$, from vertices $i$ and $(i+1)$, and from vertices succeeding the vertex $(i+1)$. First, contributions from vertices preceding vertex $i$ can be of two types, either $(A_j, \underline{A}_i)$ or $(B_j, \underline{A}_i)$ (where the underline denotes the location of the brane). When we move the brane to the $(i+1)$'th vertex these two types change into $(A_j, \underline{A}_{i+1})$ or $(B_j, \underline{A}_{i+1})$ respectively. From the table 1 it follows that this results in extending the string of Kähler parameters from $Q_{j,i}$ to $Q_{j,i+1} = Q_{j,i}Q_i$. Similarly, contributions from vertices succeeding the $(i+1)$ vertex simply change corresponding Kähler parameters by removing the factor $Q_i$. Therefore, apart from rescaling of some Kähler parameters, the structure of the part of the A-polynomial coming from these vertices does not change.

Consider now the pairing of the two vertices involved in the brane movement. The original contribution $(\underline{A}_i, A_{i+1})$, which is of type $\beta$ with parameter $Q_i$, after the movement changes to $(A_i, \underline{A}_{i+1})$ which is of type $\delta$, again with $Q_i$. This changes the structure of the A-polynomial: the factors $(1-\widehat{y}_i)(1-q^{-1}Q_i\widehat{y}_i)$ get replaced by $(1-\widehat{y}_{i+1})(Q_i-\widehat{y}_{i+1})$. This implies the transformation $\widehat{y}_{i+1} = q^{-1}Q_i\widehat{y}_i$, so that we find

$$(1-\widehat{y}_i)(1-q^{-1}Q_i\widehat{y}_i) = qQ_i^{-1}(q^{-1}Q_i-\widehat{y}_{i+1})(1-\widehat{y}_{i+1}) \sim (1-\widehat{y}_{i+1})(q^{-1}Q_i-\widehat{y}_{i+1}), \tag{75}$$

where in the last step we commuted the two factors and dropped an irrelevant constant. In the resulting A-polynomial Kähler parameters are rescaled by a factor $q^{-1}$, while rescaling $\widehat{y}_i$ introduces an extra factors $Q_i$ in the $\widehat{x}$-dependent part of the A-polynomial. These factors can be absorbed by a redefinition $\widehat{x}_{i+1} = Q_i^{f+1+d-c}\widehat{x}_i$. Ultimately, the transformation between $(\widehat{y}_i, \widehat{x}_i)$ and $(\widehat{y}_{i+1}, \widehat{x}_{i+1})$ involves only rescaling of variables, which in the language of $SL(2,\mathbb{Z})$ corresponds to the unit operator.

Consider now the case when the $(i+1)$'th vertex is of type $B$. Contributions from vertices of type $A$ preceding the $i$'th vertex change from type $\delta$ to $\alpha$, whereas from vertices of type $B$ change from $\gamma$ to $\beta$. Corresponding Kähler parameters are again rescaled by $Q_i$. Similarly, contributions of type $\beta$ from vertices succeeding the $(i+1)$'th vertex change into $\gamma$, and those

of type $\alpha$ change into $\delta$, and Kähler parameters are divided by $Q_i$. The contribution from the vertices involved in the brane movement does not change. Overall, we see that in the A-polynomial contributions of type $\beta$ and $\gamma$, as well as $\alpha$ and $\delta$, are exchanged. To achieve such a transformation by a change of variables, the operators must be related as $\widehat{y}_{i+1} \sim \widehat{y}_i^{-1}$ and $\widehat{x}_{i+1} \sim \widehat{x}_i^{-1}$ (possibly with extra rescalings, and additionally Kähler parameters before and after the transformation must be matched). Such operation is captured by the transformation $S^2$ from the point of view of $SL(2,\mathbb{Z})$.

In two other situations, when the $i$'th vertex is of type $B$, analogous analysis reveals that when the succeeding vertex is of type $A$ the transformation is $S^2$, whereas it is the identity when the vertex is of type $B$.

To sum up, moving a brane one vertex ahead in a generic strip geometry is captured by the identity or $S^2$. Moving the brane further is then captured by combining these two transformations, which overall still results in the identity or $S^2$ transformation.

## 4.3 Moving within the vertex

In turn, we analyze moving a brane with the same vertex (thus necessarily the first or the last vertex) in a strip geometry. Consider first a brane of type $A$ attached to the first vertex. If we move the brane to the other leg, the Kähler parameters remain intact, while the type of the vertex – and thus all the pairings – change. It follows from table 1 that such a transformation changes $\alpha$ contributions to $\gamma$, and $\beta$ contributions to $\delta$, and the two corresponding A-polynomials read

$$
\begin{aligned}
\widehat{A}_A(\widehat{x}_A, \widehat{y}_A) &= (1 - \widehat{y}_A) \prod_{i=1}^{s} (1 - q^{-1}\beta_i \widehat{y}_A) + (-1)^f \widehat{x}_A \prod_{i=1}^{r} (1 - \alpha_i \widehat{y}_A) \widehat{y}_A^{f+1}, \\
\widehat{A}_B(\widehat{x}_B, \widehat{y}_B) &= (1 - \widehat{y}_B) \prod_{i=1}^{s} (\delta_i - q^{-1}\widehat{y}_B) + (-1)^{f-s+r} \widehat{x}_B \prod_{i=1}^{r} (\gamma_i - \widehat{y}_B) \widehat{y}_B^{f+1+r-s}.
\end{aligned}
\tag{76}
$$

The transformation between variables and identification of the Kähler parameters take form

$$
\widehat{x}_A = \widehat{x}_B \widehat{y}_B^{r-s} \cdot (-1)^{r-s} \frac{\prod_{i=1}^{r} \gamma_i}{\prod_{i=1}^{s} \delta_i}, \qquad \widehat{y}_A = \widehat{y}_B, \qquad \beta_i = \delta_i^{-1}, \qquad \alpha_i = \gamma_i^{-1}.
\tag{77}
$$

Up to this identification of the Kähler parameters the two wave functions are thus simply related by $T^{r-s}$. This can be seen explicitly also from the transformation of the wave function

$$
\psi_{f,A}(x_A) = \sum_{n=0}^{\infty} \left((-1)^n q^{n(n-1)/2}\right)^{f+1} \frac{x_A^n}{(q,q)_n} \frac{(\alpha_1, q)_n \dots (\alpha_a, q)_n}{(\beta_1, q)_n \dots (\beta_b, q)_n},
\tag{78}
$$

$$
\psi_{f,B}(x_B) = \sum_{n=0}^{\infty} \left((-1)^n q^{n(n-1)/2}\right)^{f+1-r+s} \frac{\left(x_B \frac{\gamma_1 \cdots \gamma_r}{\delta_1 \cdots \delta_s}\right)^n}{(q,q)_n} \frac{(\gamma_1^{-1}, q)_n \dots (\gamma_c^{-1}, q)_n}{(\delta_1^{-1}, q)_n \dots (\delta_d^{-1}, q)_n},
\tag{79}
$$

which is nothing but a change of framing, indeed represented by $T^{r-s}$. This transformation does not change the underlying quiver.

## 5 Examples

In this section we illustrate earlier considerations in a number of examples.

### 5.1 $\mathbb{C}^3$

A diagram for $\mathbb{C}^3$ geometry is shown in fig. 4. The partition function for a brane attached to any of the external legs takes form

$$\psi(x) = \sum_{n=0}^{\infty} \left((-1)^n q^{n(n-1)/2}\right)^{f+1} \frac{x^n}{(q;q)_n} = P_C(q^{-(f+1)/2}x), \tag{80}$$

and can be identified as a generating function (16) associated to a one-vertex quiver encoded in the matrix $C = (f+1)$. The partition function for an antibrane takes an analogous form

$$\psi^*(x) = \sum_{n=0}^{\infty} \left((-1)^n q^{n(n-1)/2}\right)^{f} \frac{x^n}{(q;q)_n} = P_C(q^{-f/2}x), \tag{81}$$

simply with $C = (f)$, i.e. the framing is changed. For $f = 0$ we simply have $\psi^*(x) = \psi(x)^{-1}$. Quantum A-polynomials annihilating these two partition functions read

$$A(\widehat{x}, \widehat{y}) = 1 - \widehat{y} + (-1)^f \widehat{x}\widehat{y}^{f+1}, \qquad A^*(\widehat{x}, \widehat{y}) = 1 - \widehat{y} + (-1)^{f-1}\widehat{x}\widehat{y}^f. \tag{82}$$

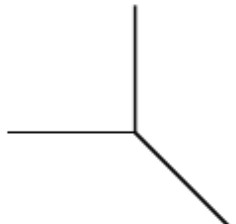

Figure 4: Toric diagram for $\mathbb{C}^3$ geometry.

For $\mathbb{C}^3$, if we change a location of a brane, its partition function does not change, so from $SL(2, \mathbb{Z})$ perspective this is just an identity operation. We can however interpret a transformation of a brane into an antibrane in this language as a power of $T$ transformation

$$\begin{pmatrix} y' \\ x' \end{pmatrix} = T^{f+1-f'} \star \begin{pmatrix} y \\ x \end{pmatrix}, \qquad T^{f+1-f'} = \begin{pmatrix} 1 & 0 \\ f+1-f' & 1 \end{pmatrix}, \tag{83}$$

so that

$$(T^{f+1-f'}\psi)(x) = \psi^*((-1)^{f+1-f'}x). \tag{84}$$

### 5.2 Resolved conifold

Let us discuss transformations of branes in the resolved conifold geometry. They were also discussed in [15] on the full quantum level, for an appropriate choice of framing. For completeness we discuss this example from our perspective, however considering also (more generally than in [15]) an arbitrary framing and invoking the relation to quivers.

For resolved conifold there are 4 external legs and thus 4 possible brane locations, whose wave-functions we denote by $\psi_i(x)$ for $i = 1, \ldots, 4$, as shown in fig. 5. These partition functions are pairwise equal, $\psi_1(x) = \psi_2(x)$ and $\psi_3(x) = \psi_4(x)$, and can be written in the

form of quiver generating functions as follows

$$\psi_1(x) = \sum_{n=0}^{\infty} \left((-1)^n q^{n(n-1)/2}\right)^{f+1} \frac{x^n}{(q;q)_n}(Q;q)_n = P_{C_1}(x, q^{-1/2}Q, Q), \qquad (85)$$

$$\psi_3(x) = \sum_{n=0}^{\infty} \left((-1)^n q^{n(n-1)/2}\right)^{f} \frac{(xQ)^n}{(q;q)_n}(Q^{-1};q)_n = P_{C_2}(xQ, q^{-1/2}Q^{-1}, Q^{-1}), \qquad (86)$$

where the corresponding quivers are encoded in matrices

$$C_1 = \begin{pmatrix} f+1 & 0 & 1 \\ 0 & 1 & 0 \\ 1 & 0 & 0 \end{pmatrix} \qquad C_2 = \begin{pmatrix} f & 0 & 1 \\ 0 & 1 & 0 \\ 1 & 0 & 0 \end{pmatrix}. \qquad (87)$$

Comparing with general notation introduced in (16), these results correspond to a choice of $\alpha_1 = Q$ for $\psi_1(x)$ and $\gamma_1 = Q$ for $\psi_3(x)$. Furthermore, we find the following the quantum A-polynomials

$$A_1(\widehat{x}, \widehat{y}) = 1 - \widehat{y} + (-1)^f \widehat{x}(1 - Q\widehat{y})\widehat{y}^{f+1}, \qquad A_3(\widehat{x}, \widehat{y}) = 1 - \widehat{y} + (-1)^{f-1}\widehat{x}(Q - \widehat{y})\widehat{y}^f. \qquad (88)$$

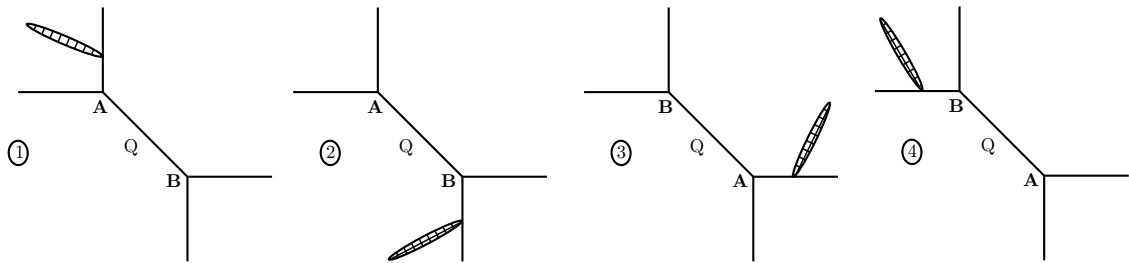

Figure 5: Toric diagram for resolved conifold.

Consider now $f = -1$. The transformation from position 1 to 3 is given by

$$\widehat{x}_3 = q\widehat{y}_1, \qquad \widehat{y}_3 = \widehat{x}_1^{-1}, \qquad f_3 = f_1 + 1, \qquad (89)$$

as can be checked by performing the transformation for the quantum A-polynomial

$$\widehat{A}_1(\widehat{x}_1, \widehat{y}_1) = 1 - \widehat{y}_1 - \widehat{x}_1(1 - Q\widehat{y}_1)\widehat{y}_1 = -\widehat{y}_3^{-1}\left(1 - \widehat{y}_3 - \widehat{x}_3(q^{-1}Q - \widehat{y}_3)\right) = -\widehat{y}_3^{-1}\widehat{A}_2(\widehat{x}_3, \widehat{y}_3), \qquad (90)$$

where the resulting quantum curve has framing $f' = 0$ and a slightly shifted Kähler parameter $q^{-1}Q$. In the classical limit this transformation is implemented by $S^{-1}$

$$\begin{pmatrix} y' \\ x' \end{pmatrix} = S^{-1} \star \begin{pmatrix} y \\ x \end{pmatrix} = \begin{pmatrix} 0 & -1 \\ 1 & 0 \end{pmatrix} \star \begin{pmatrix} y \\ x \end{pmatrix} = \begin{pmatrix} x^{-1} \\ y \end{pmatrix}. \qquad (91)$$

We can confirm this also by the analysis of classical actions. For $\psi_1(x_1)$ in framing $f_1 = -1$ the action takes form

$$S_1 = -\text{Li}_2(1 - y_1) - \ln y_1 \ln(1 - Qy_1) - \text{Li}_2(Qy_1) - \text{Li}_2(Q). \qquad (92)$$

The transformed action, with $y_3(x_3) = \frac{1 - Qx_3}{1 - x_3}$, $a = d = 0$, $c = 1$, $x_3 = y_1$, $y_3 = 1/x_1$, and using (15), takes form

$$S_1' = S_1 - \ln x_1 \ln x_3 = -\text{Li}_2(1 - x_3) - \ln x' \ln(1 - Qx_3) - \text{Li}_2(Qx_3) + \text{Li}_2(Q) + \ln y_3 \ln x_3 =$$
$$= -\ln x_3 \ln(1 - x_3) - \text{Li}_2(1 - x_3) - \text{Li}_2(Qx_3) + \text{Li}_2(Q) = \text{Li}_2(x_3) - \text{Li}_2(Qx_3) + \text{Li}_2(Q). \qquad (93)$$

Let us compare this $S_1'$ with the action $S_3$ for $\psi_2(x)$ in framing $f_3 = 0$. Using dilogarithm identities and the relation $\frac{1-y_3}{1-y_3/Q} = Qx_3$ we get

$$
\begin{aligned}
S_3 &= -\ln y_3 \ln(1 - Q^{-1}y_3) - \text{Li}_2(1 - y_3) - \text{Li}_2(Q^{-1}y_3) + \text{Li}_2(Q^{-1}) = \\
&= -\text{Li}_2\left(\frac{1}{Q}\right) - \text{Li}_2\left(\frac{1-y_3}{1-y_3/Q}\right) + \text{Li}_2\left(\frac{1}{Q}\frac{1-y_3}{1-y_3/Q}\right) + \text{Li}_2(Q^{-1}) = \\
&= -\text{Li}_2(Qx_3) + \text{Li}_2(x_3).
\end{aligned}
\tag{94}
$$

Comparing with the equation for $S_1'$ it indeed follows that $(S^{-1}\psi_1)(x) = e^{\text{Li}_2(x)/\hbar}\psi_3(x)$.

In turn, we consider the effect of $S$ transformation $\binom{y_3}{x_3} = \binom{x_1}{y_1^{-1}}$. The transformed action $(a = d = 0, c = -1)$, using $y_3(x_3) = \frac{1-x_3}{Q-x_3} = \frac{1-1/x_3}{1-Q/x_3}$, takes form

$$
\begin{aligned}
S_1' &= S_1 + \ln x_1 \ln x_3 = \ln x_3 \ln(1 - Q/x_3) - \text{Li}_2(1 - 1/x_3) - \text{Li}_2(Q/x_3) + \text{Li}_2(Q) + \ln y_3 \ln x_3 \\
&= \text{Li}_2(1/x_3) - \text{Li}_2(Q/x_3) + \text{Li}_2(Q) - \text{Li}_2(1).
\end{aligned}
\tag{95}
$$

Comparing this expression with $S_3$, we have $S_1'(x) \sim S_3(1/x)$ or $S\psi_1(x) = e^{\Delta S/\hbar}\psi_3(1/x)$. We also know that $S^2\psi_1(x) \sim \psi_1(1/x)$. As a consistency check, we can write $S^{-1} = SS^2$, which yields

$$
S^{-1}\psi_1(x) = SS^2\psi_1(x) \sim S\psi_1(1/x) \sim \psi_2(x).
\tag{96}
$$

## 5.3 Resolution of $\mathbb{C}^3/\mathbb{Z}_2$

The next example we consider is a resolution of $\mathbb{C}^3/\mathbb{Z}_2$, see fig. 6. Analogously as in the conifold case, partition functions for branes attached to external legs of a toric diagram for $\mathbb{C}^3/\mathbb{Z}_2$ are pairwise equal, and their two independent forms can be written in the quiver form

$$
\psi_1(x) = P_{C_1}(q^{-1}x, q^{-1/2}Q, Q), \qquad \psi_2(x) = P_{C_2}(q^{-1}xQ^{-1}, q^{1/2}Q^{-1}, Q^{-1}),
\tag{97}
$$

with

$$
C_1 = \begin{pmatrix} f+1 & 1 & 0 \\ 1 & 1 & 0 \\ 0 & 0 & 0 \end{pmatrix}, \qquad C_2 = \begin{pmatrix} f & 1 & 0 \\ 1 & 1 & 0 \\ 0 & 0 & 0 \end{pmatrix},
\tag{98}
$$

which corresponds to $\beta = Q$ and $\delta = Q$ respectively in our earlier notation. The corresponding classical A-polynomials are

$$
A_1(x, y) = (1-y)(1-Qy) + (-1)^f xy^{f+1}, \qquad A_2(x, y) = (1-y)(Q-y) + (-1)^{f-1}xy^f.
\tag{99}
$$

The transformation from position 1 to 2 corresponds to the following identification

$$
x_1 = Q^{f_1}x_2, \qquad y_1 = Q^{-1}y_2, \qquad f_1 = f_2 - 1.
\tag{100}
$$

Under this transformation A-polynomials transform as

$$
\begin{aligned}
A_1(x_1, y_1) &= (1-y_1)(1-Qy_1) + (-1)^f x_1 y_1^{f_1+1} = \\
&= Q^{-1}\left((1-y_2)(1-Qy_2) + (-1)^{f_2-1}x_2 y_2^{f_2}\right) = A_2(x_2, y_2),
\end{aligned}
\tag{101}
$$

which represents the identity transformation in $SL(2, \mathbb{Z})$, up to rescaling of variables by the Kähler parameter $Q$. The transformation (100) can be also written as the identity $y_1(x_1) = Q^{-1}$

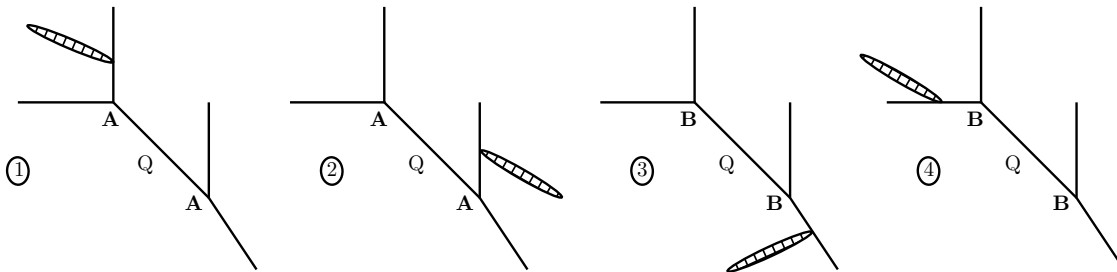

Figure 6: Toric diagram for a resolution of $\mathbb{C}^3/\mathbb{Z}_2$.

$y_2(Q^{-f}x_1)$ with $y_1(x)$ being solution to $A_1(x_1, y_1) = 0$ with framing $f_1$, and $y'(x')$ being a solution to $A_2(x_2, y_2) = 0$ with framing $f_2 = f_1 + 1$. The classical actions for the two cases above are

$$S_1(x_1) = -\frac{f_1+1}{2}(\ln y_1)^2 + \ln y_1 \ln(1 - Qy_1) - \text{Li}_2(1 - y_1) + \text{Li}_2(Qy_1) - \text{Li}_2(Q), \tag{102}$$

$$S_2(x_2) = -\frac{f_2}{2}(\ln y_2)^2 + \ln y_2 \ln(1 - Q^{-1}y_2) - \text{Li}_2(1 - y_2) + \text{Li}_2(Q^{-1}y_2) - \text{Li}_2(Q^{-1}), \tag{103}$$

and the kernel that implements the transformation from position 1 to 2 is

$$K(x_2, x_1) = \exp\Big(\frac{1}{2c\hbar}\Big(\ln\frac{x_1}{Q_1^f x_2}\Big)^2 + \frac{1}{\hbar}\ln x_2 \ln Q\Big). \tag{104}$$

In the limit $c \to 0$, this sets that $x_1 = x_1(x_2) = Q^{f_1}x_2$. The transformed action is

$$S_1'(x_2) = S_1(x_1(x_2)) + \ln x_2 \ln Q. \tag{105}$$

Again, using that $S_1$ depends on $x_1$ only through $y_1$, the relation $y_1(x_1(x_2)) = Q^{-1}y_2(x_2)$, dilogarithm identities, and $A_2(x_2, y_2) = 0$ we find

$$
\begin{aligned}
S_1'(x_2) &= -\frac{f_2}{2}\Big(\ln\frac{y_2}{Q}\Big)^2 + \ln\frac{y_2}{Q}\ln(1 - y_2) - \text{Li}_2\Big(1 - \frac{y_2}{Q}\Big) + \text{Li}_2(y_2) - \text{Li}_2(Q) + \ln x_2 \ln Q = \\
&= -\frac{f_2}{2}\big(\ln Q^{-1}y_2\big)^2 - \ln Q\ln\frac{(1 - y_2)(1 - Q^{-1}y_2)}{x_2} + \text{Li}(Q^{-1}y_2) + \\
&\quad + \ln y_2 \ln(1 - Q^{-1}y_2) - \text{Li}(1 - y_2) - \text{Li}_2(Q) = \\
&= S_2(x) - \frac{f_2}{2}(\ln Q)^2 - \text{Li}_2(Q) + \text{Li}(Q^{-1}) - \ln Q\ln(-1)^{f_2}.
\end{aligned}
$$

Therefore

$$(K\psi_1)(x) \sim \exp\Big(\frac{1}{\hbar}\Big(-\frac{f_2}{2}(\ln Q)^2 - \text{Li}_2(Q) + \text{Li}(Q^{-1}) - \ln Q\ln(-1)^{f_2}\Big)\Big)\psi_2(x). \tag{106}$$

## 5.4 Toric manifold with two Kähler parameters

As one other non-trivial example we consider a toric manifold with two Kähler parameters, captured by a diagram shown in fig. 7. There are 3 inequivalent brane positions, whose

wave-functions can be written in a quiver form respectively as

$$
\begin{aligned}
\psi_1(x) &= \sum_{n=0}^{\infty} \left((-1)^n q^{n(n-1)/2}\right)^{f+1} \frac{x^n}{(q;q)_n} \frac{(Q_1,q)_n}{(Q_1 Q_2,q)_n} = \\
&= P_{C_A}(q^{-(f+1)/2}x, q^{-1/2}Q_1, Q_1, q^{-1/2}Q_1 Q_2, Q_1 Q_2), \\
\psi_2(x) &= \sum_{n=0}^{\infty} \left((-1)^n q^{n(n-1)/2}\right)^{f} \frac{(Q_2 x)^n}{(q;q)_n} (Q_1,q)_n (Q_2^{-1},q)_n = \\
&= P_{C_B}(q^{-(f-1)/2}x Q_2, q^{-1/2}Q_1, Q_1, q^{-1/2}Q_2^{-1}, Q_2^{-1}), \\
\psi_3(x) &= \sum_{n=0}^{\infty} \left((-1)^n q^{n(n-1)/2}\right)^{f+1} \frac{x^n}{(q;q)_n} \frac{(Q_2,q^{-1})_n}{(Q_1 Q_2,q^{-1})_n} = \\
&= P_{C_A}(q^{-(f+1)/2}x Q_1^{-1}, q^{-1/2}Q_2^{-1}, Q_2^{-1}, q^{-1/2}Q_1^{-1}Q_2^{-1}, Q_1^{-1}Q_2^{-1}),
\end{aligned}
\tag{107}
$$

for quivers

$$
C_A = \begin{pmatrix} f+1 & 0 & 1 & 1 & 0 \\ 0 & 1 & 0 & 0 & 0 \\ 1 & 0 & 0 & 0 & 0 \\ 1 & 0 & 0 & 1 & 0 \\ 0 & 0 & 0 & 0 & 0 \end{pmatrix}, \qquad C_B = \begin{pmatrix} f & 0 & 1 & 0 & 1 \\ 0 & 1 & 0 & 0 & 0 \\ 1 & 0 & 0 & 0 & 0 \\ 0 & 0 & 0 & 1 & 0 \\ 1 & 0 & 0 & 0 & 0 \end{pmatrix}.
\tag{108}
$$

For $\psi_1(x)$ we identify parameters as $\alpha = Q_1$ and $\beta = Q_1 Q_2$, for $\psi_2(x)$ we identify $\alpha = Q_1$ and $\gamma = Q_2$, and for $\psi_3(x)$ we identify $\gamma = Q_2$ and $\delta = Q_1 Q_2$. The wave-functions for branes at two other locations are related to those above as $\psi_4(x) \leftrightarrow \psi_1(x)$ and $\psi_5(x) \leftrightarrow \psi_3(x)$, in addition with exchanging $Q_1 \leftrightarrow Q_2$.

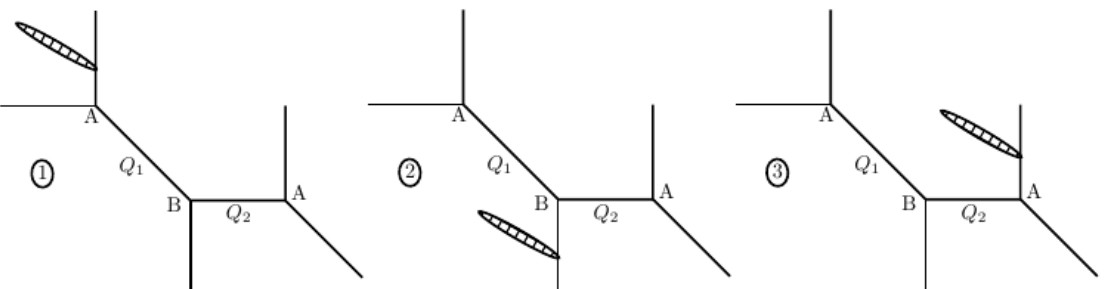

Figure 7: Toric diagram for a toric manifold with two Kähler parameters.

The classical actions $S_i$ for $\psi_i(x)$ are

$$
\begin{aligned}
S_1(x) &= -\frac{f+1}{2}(\ln y)^2 - \mathrm{Li}_2(1-y) - \ln y \ln(1-Q_1 y) + \ln y \ln(1-Q_1 Q_2 y) + \\
&\quad - \mathrm{Li}_2(Q_1 y) + \mathrm{Li}_2(Q_1) + \mathrm{Li}_2(Q_1 Q_2 y) - \mathrm{Li}_2(Q_1 Q_2), \\
S_2(x) &= -\frac{f}{2}(\ln y)^2 - \mathrm{Li}_2(1-y) - \ln y \ln(1-Q_1 y) - \ln y \ln(1-Q_2^{-1} y) + \\
&\quad - \mathrm{Li}_2(Q_1 y) + \mathrm{Li}_2(Q_1) - \mathrm{Li}_2(Q_2^{-1} y) + \mathrm{Li}_2(Q_2^{-1}), \\
S_3(x) &= -\frac{f+1}{2}(\ln y)^2 - \mathrm{Li}_2(1-y) - \ln y \ln(1-Q_2^{-1} y) + \ln y \ln(1-Q_1^{-1}Q_2^{-1} y) + \\
&\quad - \mathrm{Li}_2(Q_2^{-1} y) + \mathrm{Li}_2(Q_2^{-1}) + \mathrm{Li}_2(Q_1^{-1}Q_2^{-1} y) - \mathrm{Li}_2(Q_1^{-1}Q_2^{-1}).
\end{aligned}
$$

Finally, A-polynomials for 3 inequivalent positions of the brane take form

$$
\begin{aligned}
A_1(x,y) &= (1-y)(1-Q_1Q_2 y) + (-1)^f x(1-Q_1 y)y^{f+1},\\
A_2(x,y) &= (1-y) + (-1)^{f-1} x(1-Q_1 y)(Q_2 - y)y^f,\\
A_3(x,y) &= (1-y)(Q_1Q_2 - y) + (-1)^f x(Q_2 - y)y^{f+1}.
\end{aligned}
\tag{109}
$$

Consider now a transformation from position 3 to 1, at $f = 0$. From the above form of A-polynomials we find the following transformation rule

$$
x_3 = Q_2^{-1} x_1, \qquad y_3 = Q_1 Q_2 y_1,
\tag{110}
$$

which involves only rescaling of variables. At the classical level this can be achieved with

$$
K(x_1, x_3) = \exp\left(\frac{1}{2c\hbar}(\ln Q_2 x_3/x_1)^2 - \frac{1}{\hbar}\ln x_1 \ln Q_1 Q_2\right).
\tag{111}
$$

From the above change of variables, the A-polynomial relation $\frac{(1-y_1)(1-Q_1Q_2 y_1)}{1-Q_1 y'} = -x_1 y_1$, and (15), we find

$$
\begin{aligned}
S_3'(x_1) &= S_3(Q_2^{-1} x_1) - \ln x_1 \ln Q_1 Q_2 = \\
&= -\frac{1}{2}(\ln Q_1 Q_2 y_1)^2 + \mathrm{Li}_2(Q_1 Q_2 y_1) + \ln y_1 \ln(1-Q_1 Q_2 y_1) - \ln y_1 \ln(1-Q_1 y_1) + \\
&\quad - \mathrm{Li}_2(Q_1 y_1) + \mathrm{Li}_2(Q_2^{-1}) - \mathrm{Li}_2(1-y_1) - \mathrm{Li}_2(Q_1^{-1} Q_2^{-1}) + \ln Q_1 Q_2 \ln(-y_1) = \\
&= S_1(x_1) - \frac{1}{2}(\ln Q_1 Q_2)^2 + \mathrm{Li}_2(\frac{1}{Q_2}) - \mathrm{Li}_2(Q_1) + \mathrm{Li}_2(Q_1 Q_2) - \mathrm{Li}_2(\frac{1}{Q_1 Q_2}) + i\pi \ln Q_1 Q_2,
\end{aligned}
$$

so that

$$
(K\psi_3)(x) \sim e^{\mathrm{const}(Q_1,Q_2)/\hbar} \psi_1(x).
\tag{112}
$$

Consider now moving the brane from position 1 to 2, with $f = 0$. This is captured by the change of variables

$$
x_1 = \frac{1}{x_2}, \qquad y_1 = \frac{1}{Q_1 y_2}.
\tag{113}
$$

Indeed

$$
\begin{aligned}
A_1(x_1, y_1) &= (1-y_1)(1-Q_1Q_2 y_1) + x_1(1-Q_1 y_1)y_1 = \\
&= \frac{1}{Q_1 x_2 y_2^2}(x'(1-Q_1 y_2)(Q_2 - y_2) - (1-y_2)) = -\frac{1}{Q_1 x_2 y_2^2} A_2(x_2, y_2).
\end{aligned}
\tag{114}
$$

The result is also in framing $f = 0$, and this transformation corresponds to $S^2$ with an additional rescaling of the variables. This can be implemented by the kernel

$$
K(x_2, x_1) = \exp\left(-\frac{1}{2c\hbar}(\ln x_2 x_1)^2 + \frac{1}{\hbar}\ln x_2 \ln Q_1\right).
\tag{115}
$$

Using $y_1(1/x_2) = Q_1^{-1} y_2^{-1}(x_2)$, some dilogarithm identities, and the A-polynomial relation $x_2 = \frac{1-y_2}{(1-Q_1 y_2)(Q_2 - y_2)}$, we find

$$
\begin{aligned}
S_1'(x_2) &= S_1(1/x_2) + \ln x_2 \ln Q_1 = \\
&= -\mathrm{Li}_2(1-y_2) - \mathrm{Li}_2(Q_1 y_2) - \mathrm{Li}_2(y_2/Q_2) - \ln y' \ln(1-Q_1 y_2) - \ln y_2 \ln(1-y_2/Q_2) \\
&\quad + \mathrm{Li}_2(Q_1) - \mathrm{Li}_2(Q_1 Q_2) + 2\mathrm{Li}_2(1) - \frac{1}{2}(\ln Q_2)^2 + i\pi \ln Q_2 = \\
&= S_2(x_2) - \mathrm{Li}_2(Q_2^{-1}) - \mathrm{Li}_2(Q_1 Q_2) + 2\mathrm{Li}_2(1) - \frac{1}{2}(\ln Q_2)^2 + i\pi \ln Q_2,
\end{aligned}
\tag{116}
$$

so that

$$(K\psi_1)(x) \sim e^{\mathrm{const}(Q_1,Q_2)/\hbar}\psi_2(x). \tag{117}$$

Finally, we consider moving the brane from position 2 to 3 with $f = 0$, which is captured by the following relations

$$A_2(x_2, y_2) = -x_3^{-1}y_3^{-2}A_3(x_3, y_3), \tag{118}$$

$$x_2 = Q_2^{-1}x_3^{-1}, \ y_2 = Q_2 y_3^{-1}. \tag{119}$$

The corresponding kernel is

$$K(x_3, x_2) = \exp\Big(-\frac{1}{2c\hbar}(\ln(Q_2 x_2 x_3))^2 + \frac{1}{\hbar}\ln Q_2 \ln x_3\Big). \tag{120}$$

By the same computation as in previous examples, one can show that the kernel describes the transformation of the action correctly

$$(K\psi_2)(x) \sim e^{\mathrm{const}(Q_1,Q_2)/\hbar}\psi_3(x). \tag{121}$$

## 5.5 Periodic chain geometry

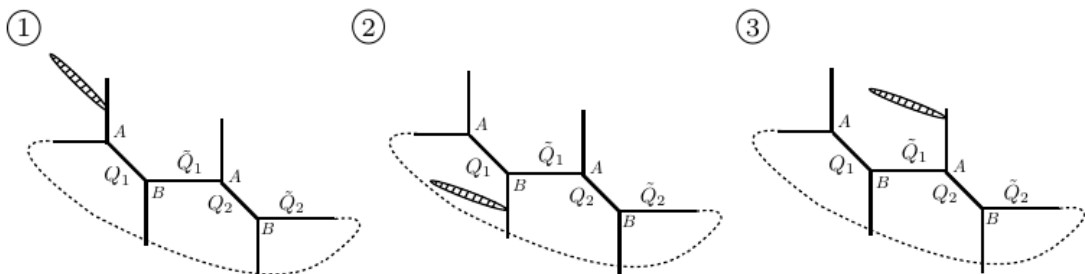

Figure 8: The periodic chain geometry with $N = 2$.

As the last example we consider transformations of branes in a manifold represented by the simplest non-trivial periodic diagram shown in fig. 8. Consider 3 wave-functions for branes located in the positions shown in this figure

$$\psi_1(x) = \sum_{n=0}^{\infty}\big((-1)^n q^{n(n-1)/2}\big)^{f+1}\frac{x^n}{(q;q)_n} \times$$
$$\times \prod_{m=1}^{\infty}\frac{(Q_1 p^{m-1};q)_n(Q_1^{-1}p^m:q^{-1})_n(Q_1\tilde{Q}_1 Q_2 p^{m-1};q)_n(Q_1^{-1}\tilde{Q}_1^{-1}Q_2^{-1}p^m:q^{-1})_n}{(qp^m;q)_n(q^{-1}p^m;q^{-1})_n(Q_1\tilde{Q}_1 p^{m-1};q)_n(\tilde{Q}_1^{-1}\tilde{Q}_1^{-1}p^m;q^{-1})_n}, \tag{122a}$$

$$\psi_2(x) = \sum_{n=0}^{\infty}\big((-1)^n q^{n(n-1)/2}\big)^{f+1}\frac{\big(Q_2^{-1}x\big)^n}{(q;q)_n} \times$$
$$\times \prod_{m=1}^{\infty}\frac{(Q_1 p^{m-1};q)_n(Q_1^{-1}p^m:q^{-1})_n(\tilde{Q}_1^{-1}p^{m-1};q)_n(\tilde{Q}_1 p^m:q^{-1})_n}{(qp^m;q)_n(q^{-1}p^m;q^{-1})_n(\tilde{Q}_1^{-1}Q_2^{-1}p^{m-1};q)_n(\tilde{Q}_1 Q_2 p^m;q^{-1})_n}, \tag{122b}$$

$$\psi_3(x) = \sum_{n=0}^{\infty}\big((-1)^n q^{n(n-1)/2}\big)^{f+1}\frac{\big(Q_1^{-1}x\big)^n}{(q;q)_n} \times$$
$$\times \prod_{m=1}^{\infty}\frac{(\tilde{Q}_1^{-1}p^{m-1};q)_n(\tilde{Q}_1 p^m:q^{-1})_n(Q_2 p^{m-1};q)_n(Q_2^{-1}p^m:q^{-1})_n}{(qp^m;q)_n(q^{-1}p^m;q^{-1})_n(Q_1^{-1}\tilde{Q}_1^{-1}p^{m-1};q)_n(Q_1\tilde{Q}_1 p^m;q^{-1})_n}. \tag{122c}$$

Note that $\psi_3(x)$ is related to $\psi_1(x)$ by exchanging Kähler parameters

$$\psi_3(x) = \psi_1(x)|_{Q_1 \leftrightarrow Q_2, \tilde{Q}_1 \leftrightarrow \tilde{Q}_2}, \tag{123}$$

as also illustrated in fig. 9.

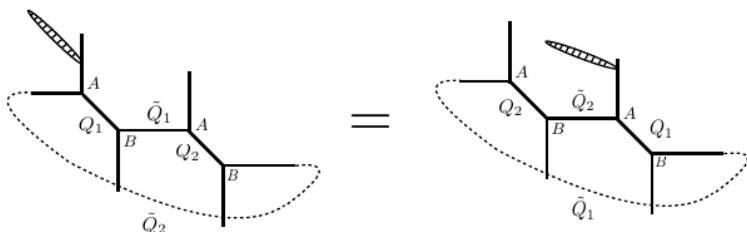

Figure 9: The relation of web diagrams ① and ③.

We find that classical mirror curves corresponding to the above wave-functions take form

$$H_1(x,y) = \theta(y)\,\theta(Q_1\tilde{Q}_1 y) + (-1)^f\, xy^{f+1}\theta(Q_1 y)\,\theta(Q_1\tilde{Q}_1 Q_2 y), \tag{124a}$$

$$H_2(x,y) = \theta(y)\,\theta(\tilde{Q}_1^{-1}Q_2^{-1}y) + Q_2^{-1}(-1)^f\, xy^{f+1}\theta(Q_1 y)\,\theta(\tilde{Q}_1^{-1}y), \tag{124b}$$

$$H_3(x,y) = \theta(y)\,\theta(Q_1^{-1}\tilde{Q}_1^{-1}y) + Q_1^{-1}(-1)^f\, xy^{f+1}\theta(\tilde{Q}_1^{-1}y)\theta(Q_2 y), \tag{124c}$$

while classical A-polynomials assigned to regularized wave-functions (as explained in section 2.4) take form

$$A_1(x,y) = (y;p)_\infty(p^{-1}y;p^{-1})_\infty(Q_1\tilde{Q}_1 y;p)_\infty(Q_1\tilde{Q}_1 p^{-1}y;p^{-1})_\infty +$$
$$+ (Q_1 Q_2)^{1/2}(-1)^f xy^{f+1}(Q_1 y;p)_\infty(Q_1 p^{-1}y;p^{-1})_\infty(Q_1\tilde{Q}_1 Q_2 y;p)_\infty(Q_1\tilde{Q}_1 Q_2 p^{-1}y;p^{-1})_\infty, \tag{125a}$$

$$A_2(x,y) = (y;p)_\infty(p^{-1}y;p^{-1})_\infty(\tilde{Q}_1^{-1}Q_2^{-1}y;p)_\infty(\tilde{Q}_1^{-1}Q_2^{-1}p^{-1}y;p^{-1})_\infty +$$
$$+ \left(Q_1 Q_2^{-1}\right)^{1/2}(-1)^f xy^{f+1}(Q_1 y;p)_\infty(Q_1 p^{-1}y;p^{-1})_\infty(\tilde{Q}_1^{-1}y;p)_\infty(\tilde{Q}_1^{-1}p^{-1}y;p^{-1})_\infty, \tag{125b}$$

$$A_3(x,y) = (y;p)_\infty(p^{-1}y;p^{-1})_\infty(Q_1^{-1}\tilde{Q}_1^{-1}y;p)_\infty(Q_1^{-1}\tilde{Q}_1^{-1}p^{-1}y;p^{-1})_\infty +$$
$$+ \left(Q_1^{-1}Q_2\right)^{1/2}(-1)^f xy^{f+1}(\tilde{Q}_1^{-1}y;p)_\infty(\tilde{Q}_1^{-1}p^{-1}y;p^{-1})_\infty(Q_2 y;p)_\infty(Q_2 p^{-1}y;p^{-1})_\infty. \tag{125c}$$

The corresponding classical actions $S_i$, for $i = 1, 2, 3$, are

$$S_i = -\frac{1}{2}(\ln y_i) - \mathrm{Li}_2(1 - y_i) - \ln y_i \ln\Big(\prod_{n=1}^\infty \frac{(1 - \alpha_n^i y_i)(1 - \gamma_n^i y_i)}{(1 - \beta_n^i y_i)(1 - \delta_n^i y_i)}\Big) +$$
$$+ \sum_{n=1}^\infty \Big( -\mathrm{Li}_2\left(\alpha_n^i y_i\right) - \mathrm{Li}_2\left(\gamma_n^i y_i\right) + \mathrm{Li}_2\left(\beta_n^i y_i\right) + \mathrm{Li}_2\left(\delta_n^i y_i\right)$$
$$+ \mathrm{Li}_2(\alpha_n) + \mathrm{Li}_2(\gamma_n) - \mathrm{Li}_2(\beta_n) - \mathrm{Li}_2(\delta_n)\Big),$$

where

$$\alpha^1 = \{Q_1 p^{n-1}, Q_1 \tilde{Q}_1 Q_2 p^{n-1}, n = 1, ..., \infty\}, \quad \beta^1 = \{p^n, Q_1 \tilde{Q}_1 p^{n-1}, n = 1, ..., \infty\},$$
$$\gamma^1 = \{Q_1 p^{-n}, Q_1 \tilde{Q}_1 Q_2 p^{-n}, n = 1, ..., \infty\}, \quad \delta^1 = \{p^{-n}, Q_1 \tilde{Q}_1 p^{-n}, n = 1, ..., \infty\}, \quad (126a)$$
$$\alpha^2 = \{Q_1 p^{n-1}, \tilde{Q}_1^{-1} p^{n-1}, n = 1, ..., \infty\}, \quad \beta^2 = \{p^n, \tilde{Q}_1^{-1} Q_2^{-1} p^{n-1}, n = 1, ..., \infty\},$$
$$\gamma^2 = \{Q_1 p^{-n}, \tilde{Q}_1^{-1} p^{-n}, n = 1, ..., \infty\}, \quad \delta^2 = \{p^{-n}, \tilde{Q}_1^{-1} Q_2^{-1} p^{-n}, n = 1, ..., \infty\}, \quad (126b)$$
$$\alpha^3 = \{\tilde{Q}_1^{-1} p^{n-1}, Q_2 p^{n-1}, n = 1, ..., \infty\}, \quad \beta^3 = \{p^n, Q_1^{-1} \tilde{Q}_1^{-1} p^n, n = 1, ..., \infty\},$$
$$\gamma^3 = \{\tilde{Q}_1^{-1} p^{-n}, Q_2 p^{-n+1}, n = 1, ..., \infty\}, \quad \delta^3 = \{p^{-n}, Q_1^{-1} \tilde{Q}_1^{-1} p^{-n}, n = 1, ..., \infty\}. \quad (126c)$$

In what follows we take advantage interchangeably of the expressions for $H(x, y)$ or $A(x, y)$.

Let us consider first the transformation from position 1 to 2. From the mirror curves $H_1(x, y)$ and $H_2(x, y)$ we find the following relations

$$H_1(x_1, y_1) = Q_1^{-1} \tilde{Q}_1 Q_2 x^{-1} y_2^{-3} H_2(x_2, y_2), \qquad x_1 = x_2^{-1}, \qquad y_1 = Q_1^{-1} y_2^{-1}. \quad (127)$$

The relation between $(x_1, y_1)$ and $(x_2, y_2)$ is the same as in a non-periodic strip geometry. The corresponding kernel takes form

$$K(x_2, x_1) = \exp\left(\frac{1}{\hbar} (\ln x_2 x_1)^2 + \frac{1}{\hbar} \ln x_2 \ln Q_1\right), \quad (128)$$

and so the transformation of the action reads

$$S_1'(x_2) = S_1(x_1(x_2)) + \ln x_2 \ln Q_1. \quad (129)$$

By using the form of A-polynomial and the following identities

$$-\sum_{n=1}^{\infty} \left(\mathrm{Li}_2(Q_2 Q_3 p^n y_2^{-1}) + \mathrm{Li}_2(Q_2 Q_3 p^{-n} y_2^{-1})\right) - \mathrm{Li}_2(Q_2 Q_3 y_2^{-1}) =$$
$$= \sum_{n=1}^{\infty} \left(\mathrm{Li}_2(Q_2^{-1} Q_3^{-1} p^{-n} y_2) + \mathrm{Li}_2(Q_2^{-1} p^n y_2)\right) + \mathrm{Li}_2(Q_2^{-1} Q_3^{-1} y_2^{-1}), \quad (130a)$$
$$-\sum_{n=1}^{\infty} \left(\mathrm{Li}_2(p^n y_2^{-1}) - \mathrm{Li}_2(Q_1^{-1} p^n y_2^{-1})\right) = \sum_{n=1}^{\infty} \left(\mathrm{Li}_2(p^{-n} y_2) - \mathrm{Li}_2(Q_1 p^{-n} y_2)\right) + \frac{1}{2} \ln Q_1 \ln y_2, \quad (130b)$$

we find

$$S_1'(x_2) = S_2(x_2) + \mathrm{const.} \quad (131)$$

Therefore, we conclude that

$$(K\psi_1)(x) \sim e^{\mathrm{const}/\hbar} \psi_2(x). \quad (132)$$

In turn, consider the transformation from position 3 to 1. The relation between variables is again the same as in a non-periodic geometry

$$H_3(x_3, y_3) = H_1(x_1, y_1), \qquad x_3 = \tilde{Q}_1^{-1} x_1, \qquad y_3 = Q_1 \tilde{Q}_1 y_1, \quad (133)$$

which leads to the same kernel

$$K(x_1, x_3) = \exp\left(\frac{1}{\hbar} \left(\ln \tilde{Q}_1 x_3 / x_1\right)^2 - \frac{1}{\hbar} \ln x_1 \ln Q_1 \tilde{Q}_1\right) \quad (134)$$

and the same transformation of the action $S_3(x_3)$

$$S_3'(x_1) = S_3(x_3(x_1)) - \ln x_1 \ln Q_1 \tilde{Q}_1. \tag{135}$$

After some calculation we find

$$S_3'(x_1) = S_1(x_1) + \text{const}, \tag{136}$$

so that

$$(K\psi_3)(x) \sim e^{\text{const}} \psi_1(x). \tag{137}$$

**Kernel as the identity operator**

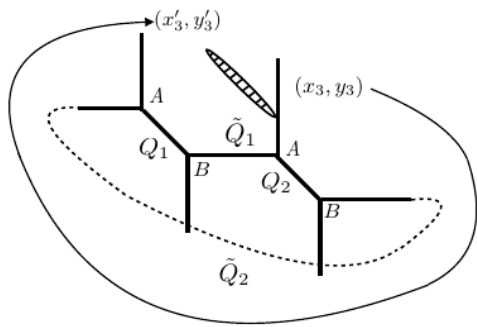

Figure 10: The relation of web diagrams ① and ③.

At the end, let us consider what happens when a brane moves around a horizontal axis, either changing positions as ① → ③ → ① or ③ → ① → ③, as shown in fig. 10. From the viewpoint of mirror curves, on one hand we have the relations of the variables $x_i$ and $y_i$ given in (133), where the brane moves through the horizontal and slanting lines with Kähler parameters $Q_1$ and $\tilde{Q}_1$. On the other hand the brane can pass through the lines with the Kähler parameters $Q_2$ and $\tilde{Q}_2$. In this case the relation of the variables $x_i, y_i$ of the mirror curves $H_i(x_i, y_i)$ reads

$$x_3 = \tilde{Q}_2 x_1, \qquad y_3 = p^{-1} Q_1 \tilde{Q}_1 y_1, \tag{138}$$

where $f = 0$. To show this relation we used periodic properties of theta functions

$$\theta(px) = -x^{-1}\theta(x), \qquad \theta(x) = -x\theta(x^{-1}). \tag{139}$$

Using (133) and (139), we find the relation of the variables of the curve $H_3$ before and after moving the brane

$$H_3(x_3, y_3) = H_3(x_3', y_3'), \qquad x_3 = pQ_1^{-1}Q_2^{-1}x_3', \qquad y_3 = p^{-1}y_3', \tag{140}$$

where we define the variables $(x_3, y_3)$ and $(x_3', y_3')$ as those of $H_3$ before and after moving the brane, respectively. From this relation we determine the kernel

$$K(x_3, x_3') = \exp\left(\frac{1}{\hbar}\left(\ln(pQ_1^{-1}Q_2^{-1}x_3/x_3')\right)^2 - \frac{1}{\hbar}\ln x_3' \ln p\right), \tag{141}$$

the transformation law of the action

$$S_3'(x_3') = S_3(x_3(x_3')) - \ln x_3' \ln p, \tag{142}$$

and ultimately, writing the constant term explicitly

$$S_3'(x_3) = S_3(x_3) - \frac{1}{2}(\ln p)^2.$$ (143)

Then, we may regard the kernel $K(x_3', x_3)$ as an identity operator up to a constant

$$K(x_3', x_3) \sim \delta(\ln x_3/x_3').$$ (144)

To understand the property (144) deeper, as a simple but non-trivial example let us consider the non-periodic geometry discussed in section 5.4, see fig. 7. The kernels $K(x_3, x_2)$, $K(x_2, x_1)$ and $K(x_1, x_3)$ are given in (120), (115) and (111), respectively. Then, the kernel describing the brane moving around the geometry is given by

$$K(x_3', x_3) = \int dx_1 dx_2 K(x_3', x_2) K(x_2, x_1) K(x_1, x_3) =$$
$$= \text{const} \times \exp\left(\frac{1}{2c\hbar}\left(-(\ln(x_3'/x_3))^2 + \mathcal{O}(c^0)\right)\right).$$ (145)

When we take $c \to 0$ limit, this function approaches zero quickly unless $x_3' = x_3$, so that $K(x_3', x_3)$ behaves like delta function, as claimed in (144). However, precisely speaking, the kernel is not delta function; the main difference is that actually the kernel does not diverge when we set $x_3 = x_3'$. Nevertheless, when we consider the integral for $x_3$ corresponding to the transformation of the wave function, the behavior is similar to the delta function.

We can make this observation a bit more precise by redefining the kernel, here for the special case of the identity operation,

$$K(x', x) = \exp\left(-\frac{1}{2c\hbar}(\ln x'/x)^2\right) \to \frac{1}{\sqrt{2c\pi\hbar}}\exp\left(-\frac{1}{2c\hbar}(\ln x'/x)^2\right).$$ (146)

This gives, in the limit $c \to 0$,

$$\int dx K(x', x)\psi(x) = \psi(x')\int dx K(x', x) = x'\psi(x'),$$ (147)

and the new kernel indeed acts like the delta function up to the extra $x'$ term.

# A  Topological vertex for strip geometries

In this appendix, following [6, 19], we summarize how the topological vertex formalism simplifies for toric strip geometries, and generalize it to include branes attached to non-vertical legs.

As mentioned in section 2.1, a toric diagram for a strip geometry takes form of a string of trivalent vertices labeled by $i = 1, \ldots, m$. Each vertex is of type $A$ or $B$, assigned as follows: the first vertex is of type $A$ if in the clockwise direction the vertical edge precedes the internal edge of the geometry (otherwise it is of type $B$); and (recursively) the next vertex is of the same (or the opposite) type as the preceding vertex if the two vertices are connected by $(-2, 0)$ (or respectively by $(-1, -1)$) line. Each vertex other than the first or the last one has one vertical leg attached that extends to infinity, and the first and the last vertex have two such legs. The total topological string amplitude for branes in such a geometry, involving both closed and open contributions, takes form

$$Z = Z^{\text{closed}}(Q) \cdot \psi^{\text{open}}(Q, x) = \sum_{\{P_i\}} Z_{\{P_i\}} \prod_i \text{Tr}_{P_i} X_i,$$ (148)

and depends on (closed) Kähler parameters $Q = \{Q_k\}$ and (open) brane moduli $x = \{x_i\}$ that are assembled into $X = \text{diag}(x_1, x_2, \dots)$. The total partition function factorizes into contributions from closed and open strings and can be computed by summing over $Z_{\{P_i\}}$ that depends on matrices $P_i$ that encode boundary conditions of a brane. The contributions for branes attached to vertical edges of a strip diagram take form

$$Z_{P_i} = \prod_i s_{P_i}(q^\rho) \prod_{i,j} \{P_i^*, P_j^*\}_{Q_{ij}}^{\pm 1}, \tag{149}$$

where $P^*$ (that denotes either $P$ or $P^T$) and powers $\pm 1$ depend on the types ($A$ or $B$) of vertices $i$ and $j$, while

$$\{P_i, P_j\} = \prod_k (1 - Q_{ij} q^k)^{C_k(P_i, P_j)} \exp\Big( \sum_{m=1}^\infty \frac{Q_{ij}^m}{m\left(2\sin\frac{m\hbar}{2}\right)^2} \Big) \tag{150}$$

are symmetric under exchanging $P_i$ and $P_j$ and depend on $Q_{ij} = Q_i Q_{i+1} \cdots Q_{j-1}$, i.e. a product of Kähler parameters $Q_k$ associated to internal legs joining the pair of vertices. The exponents $C_k(P,R)$ are defined by

$$\sum_k C_k(P,R) q^k = \frac{q}{(q-1)^2}\Big(1 + (q-1)\sum_{i=1}^{d_P} q^{-1} \sum_{j=0}^{P_i-1} q^j\Big)\Big(1 + (q-1)\sum_{i=1}^{d_R} q^{-1} \sum_{j=0}^{R_i-1} q^j\Big) - \frac{q}{(q-1)^2}. \tag{151}$$

For a pair of vertices of types $(A_i, A_j)$, $(A_i, B_j)$, $(B_i, A_j)$, $(B_i, B_j)$ the corresponding factors in (149) are $\{P_i, P_j^T\}^{-1}$, $\{P_i, P_j\}$, $\{P_i^T, P_j^T\}$, $\{P_i^T, P_j\}^{-1}$ respectively.

We are concerned with situations when there is only one brane. In this case $x$ is a single variable and $\text{Tr}_P x \neq 0$ only for symmetric representations $P = S^n$ and $\text{Tr}_{S^n}(x) = x^n$. In this case the factors (150) for symmetric and empty representation take form

$$\{(n), \bullet\}_Q = (Q; q)_n \{\bullet, \bullet\}_Q, \qquad \{(n)^T, \bullet\}_Q = (Q; q^{-1})_n \{\bullet, \bullet\}_Q, \tag{152}$$

with the closed string contribution

$$\{\bullet, \bullet\}_Q = \exp\Big( \sum_{m=1}^\infty \frac{Q_{ij}^m}{m\left(2\sin\frac{m\hbar}{2}\right)^2} \Big). \tag{153}$$

The open string partition function, with a single brane at the $i$-th vertex in the framing $f$, takes then the form

$$\psi_{f,i}(x) = \sum_{n=0}^\infty \big((-1)^n q^{n(n-1)/2}\big)^{f+1} \frac{x^n}{(q;q)_n} \prod_{j<i} X_{ji} \prod_{j>i} X_{ij}, \tag{154}$$

where $X_{ij}$ are given in table 2.

## A.1 Topological vertex and branes on vertical legs

In what follows we show how the above rules generalize to the situation when a brane is attached to a horizontal leg of the first or the last vertex of a strip. To this end we highlight the crucial steps in the derivation of the above rules and adapt to more general situations. We start with recalling the formalism of the topological vertex, in particular gluing of two vertices into two prototypical geometries $(-2, 0)$ and $(-1, -1)$.

The topological vertex amplitude in the canonical framing takes form

$$C_{\lambda\mu\nu} = q^{\kappa(\lambda)/2} s_\nu(q^\rho) \sum_\eta s_{\lambda^T/\eta}(q^{\nu+\rho}) s_{\mu/\eta}(q^{\nu^T+\rho}), \tag{155}$$

Table 2: The rules for assigning the contribution $X_{ij}$ to the open string partition function in a strip geometry with a single brane placed on the vertical external leg of the $i$-th vertex and in symmetric representation $n$. A position of the brane in the pairing is denoted by an underline.

| $X_{ij}$, $i < j$ | $X_{ji}$, $i > j$ |
|---|---|
| $(\underline{A}, B) \to \{(n), \bullet\}_{Q_{ij}} \to (Q_{ij}; q)_n$ | $(A, \underline{B}) \to \{\bullet, (n)\}_{Q_{ij}} \to (Q_{ji}; q)_n$ |
| $(\underline{B}, A) \to \{(n)^T, \bullet\}_{Q_{ij}} \to (Q_{ij}; q^{-1})_n$ | $(B, \underline{A}) \to \{\bullet, (n)^T\}_{Q_{ij}} \to (Q_{ji}; q^{-1})_n$ |
| $(\underline{A}, A) \to \{(n), \bullet\}_{Q_{ij}}^{-1} \to (Q_{ij}; q)_n^{-1}$ | $(B, \underline{B}) \to \{\bullet, (n)\}_{Q_{ij}}^{-1} \to (Q_{ji}; q)_n^{-1}$ |
| $(\underline{B}, B) \to \{(n)^T, \bullet\}_{Q_{ij}}^{-1} \to (Q_{ij}; q^{-1})_n^{-1}$ | $(A, \underline{A}) \to \{\bullet, (n)^T\}_{Q_{ij}}^{-1} \to (Q_{ji}; q^{-1})_n^{-1}$ |

where $\lambda$, $\mu$ and $\nu$ are Young diagrams, $q^{\nu+\rho} \equiv (q^{\nu_1-1/2}, q^{\nu_2-3/2}, q^{\nu_3-5/2}, \dots)$, a superscript $T$ denotes transposition, and

$$\kappa_\lambda = |\lambda| + \sum_i \lambda_i(\lambda_i - 2i) = -\kappa_{\lambda^T}, \qquad |\lambda| = \sum_i \lambda_i. \tag{156}$$

$C_{\lambda\mu\nu}$ is symmetric under cyclic permutations of indices. For $n_i$ denoting the framing change of edge $v_i$ with respect to the canonical framing $f_i$, the vertex amplitude transforms as

$$C_{\alpha_1\alpha_2\alpha_3}^{f_1-n_1v_1, f_2-n_2v_2, f_3-n_3v_3} = (-1)^{\sum_i n_i|\alpha_i|} q^{\sum_i n_i \kappa(\alpha_i)/2} C_{\alpha_1\alpha_2\alpha_3}^{f_1,f_2,f_3}. \tag{157}$$

Here $f_i$ and $v_i$ are two-dimensional integer vectors such that $f_i \wedge v_i = 1$. The canonical framing is then $f_i = v_{i-1}$, see fig. 11. When gluing vertices, their framings must be opposite. For illustration, consider gluing vertices into local geometries of type $(-2, 0)$ and $(-1, -1)$.

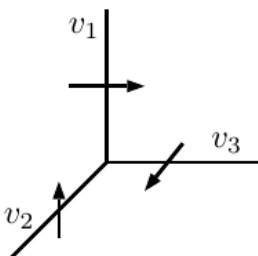

Figure 11: The topological vertex in the canonical framing.

**Gluing of $(-2, 0)$ geometry**

A diagram for $(-2, 0)$ geometry is shown in fig. 12; in this case the partition function reads

$$C_{\beta_1,\beta_2,\gamma_2,\gamma_1}^{(-2,0)} = \sum_\alpha C_{\alpha\gamma_1\beta_1}^{f_\alpha} C_{\gamma_2\alpha^T\beta_2}^{f'_{\alpha^T}} (-1)^{|\alpha|} Q^{|\alpha|}. \tag{158}$$

The gluing condition requires that $f_\alpha = -f'_{\alpha^T}$ where $f_\alpha = v_{\gamma_1} - nv_\alpha$ and $f'_{\alpha^T} = v_{\beta_2} - n'v'_\alpha$. From the geometry we have $v'_\alpha = -v_\alpha$, so that the gluing condition implies $v_{\gamma_1} - (n-n')v_\alpha = -v_{\beta_2}$. Using $v_{\gamma_1} \wedge v_\alpha = (-1,-1) \wedge (1,0) = 1$, we solve for the relative framing $n - n' = v_{\gamma_1} \wedge (v_{\gamma_1} + v_{\beta_2}) =$

$v_{\gamma_1} \wedge v_{\beta_2} = -1$. This implies that framings differ by $-1$, so that (158) takes form

$$
C^{(-2,0)}_{\beta_1,\beta_2,\gamma_2,\gamma_1} = \sum_\alpha C_{\alpha\gamma_1\beta_1} C_{\gamma_2\alpha^T\beta_2} Q^{|\alpha|} q^{-\kappa(\alpha)/2} = s_{\beta_1}(q^\rho) s_{\beta_2}(q^\rho) [\beta_1\beta_2^T]_Q q^{\kappa(\gamma_2)/2} \times
$$
$$
\times \sum_{\eta_1,\eta_2} s_{\gamma_1/\eta_1}(q^{\rho+\beta_1^T}) s_{\gamma_2^T/\eta_2}(q^{\rho+\beta_2}) \sum_\kappa s_{\eta_2/\kappa}(q^{\rho+\beta_1}) s_{\eta_1/\kappa}(q^{\rho+\beta_2^T}) Q^{|\eta_1|+|\eta_2|-|\kappa|}.
$$
$$(159)$$

For $\gamma_1 = \bullet$ this expression reduces to

$$
C^{(-2,0)}_{\beta_1,\beta_2,\gamma_2,\bullet} = s_{\beta_1}(q^\rho) s_{\beta_2}(q^\rho) [\beta_1\beta_2^T]_Q q^{\kappa(\gamma_2)/2} \sum_{\eta_2} s_{\gamma_2^T/\eta_2}(q^{\rho+\beta_2}) s_{\eta_2}(q^{\rho+\beta_1}) Q^{|\eta_2|}. \tag{160}
$$

The framings of the outer edges are in principal arbitrary. The derivation in [19] assumed that branes can be placed only on $\beta_i$ edges, while $\gamma_i$'s are trivial or are summed over upon gluing.

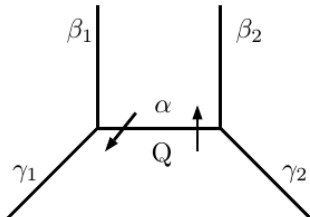

Figure 12: $(-2,0)$ curve with both inner edges in the canonical framing.

Consider now placing the brane on a horizontal line, i.e. assuming nontrivial $\gamma_1$ or $\gamma_2$. In the first case, we rewrite (159) by cycling permutation of the first vertex

$$
C^{(-2,0)}_{\beta_1,\beta_2,\gamma_2,\gamma_1} = \sum_\alpha C_{\beta_1\alpha\gamma_1} C_{\gamma_2\alpha^T\beta_2} Q^{|\alpha|} q^{-\kappa(\alpha)/2} = q^{\kappa(\beta_1)/2} q^{\kappa(\gamma_2)/2} s_{\gamma_1}(q^\rho) s_{\beta_2}(q^\rho) \times
$$
$$
\times \sum_{\eta_1,\eta_2} s_{\beta_1^T/\eta_1}(q^{\gamma_1+\rho}) s_{\gamma_2^T/\eta_2}(q^{\beta_2+\rho}) \sum_\alpha s_{\alpha/\eta_1}(q^{\gamma_1^T+\rho}) s_{\alpha/\eta_2}(q^{\beta_1^T+\rho}) Q^{|\alpha|} q^{-\kappa(\alpha)/2}.
$$

For $\beta_1 = \beta_2 = \bullet$ we also have $\eta_1 = \bullet$ and the above partition function simplifies to

$$
C^{(-2,0)}_{\bullet,\bullet,\gamma_2,\gamma_1} = q^{\kappa(\gamma_2)/2} s_{\gamma_1}(q^\rho) \sum_{\eta_2} s_{\gamma_2^T/\eta_2}(q^\rho) \sum_\alpha s_\alpha(q^{\gamma_1^T+\rho}) s_{\alpha^T/\eta_2}(q^\rho) Q^{|\alpha|} q^{-\kappa(\alpha)/2} =
$$
$$
= q^{\kappa(\gamma_2)/2} s_{\gamma_1}(q^\rho) [\gamma_1^T, \bullet]_Q \sum_{\eta_2} s_{\gamma_2^T/\eta_2}(q^\rho) s_{\eta_2}(q^{\gamma_1^T+\rho}) Q^{|\eta_2|},
$$
$$(161)$$

where we used identities $s_\alpha(x) = q^{\kappa(\alpha)/2} s_{\alpha^T}(x)$ and $c^{|\alpha|} s_\alpha(x) = s_\alpha(cx)$ and the formula

$$
\sum_\alpha s_{\alpha/\eta_1}(x) s_{\alpha/\eta_2}(y) = \prod_{i,j} (1-x_iy_j)^{-1} \sum_\kappa s_{\eta_2/\kappa}(x) s_{\eta_1/\kappa}(y). \tag{162}
$$

Consider now a brane placed on $\gamma_2$. By cyclic permutation of the second vertex in (159)

$$
C^{(-2,0)}_{\beta_1,\beta_2,\gamma_2,\gamma_1} = \sum_\alpha C_{\alpha\gamma_1\beta_1} C_{\alpha^T\beta_2\gamma_2} Q^{|\alpha|} q^{-\kappa(\alpha)/2} = s_{\beta_1}(q^\rho) s_{\gamma_2}(q^\rho) \times
$$
$$
\times \sum_{\eta_1,\eta_2} s_{\gamma_1/\eta_1}(q^{\beta_1^T+\rho}) s_{\beta_2/\eta_2}(q^{\gamma_2^T+\rho}) s_{\alpha^T/\eta_1}(q^{\beta_1+\rho}) s_{\alpha/\eta_2}(q^{\gamma_2+\rho}) Q^{|\alpha|} q^{-\kappa(\alpha)/2}.
$$

We set now $\beta_1 = \beta_2 = \bullet$, which also imposes $\eta_2 = \bullet$ and yields

$$
\begin{aligned}
C^{(-2,0)}_{\bullet,\bullet,\gamma_2,\gamma_1} &= s_{\gamma_2}(q^\rho) \sum_{\eta_1} s_{\gamma_1/\eta_1}(q^\rho) \sum_\alpha s_{\alpha^T/\eta_1}(q^\rho) s_\alpha(q^{\gamma_2+\rho}) Q^{|\alpha|} q^{-\kappa(\alpha)/2} = \\
&= s_{\gamma_2}(q^\rho) [\bullet, \gamma_2]_Q \sum_{\eta_1} s_{\gamma_1/\eta_1}(q^\rho) s_{\eta_1}(q^{\gamma_2+\rho}) Q^{|\eta_1|}.
\end{aligned}
\tag{163}
$$

We can now consider different scenarios. We are interested in situations with a single brane. When it is attached to the first vertex, there are two possible configurations

$$
\begin{aligned}
C^{(-2,0)}_{\beta_1,\bullet,\gamma_2,\bullet} &= q^{\kappa(\gamma_2)/2} s_{\beta_1}(q^\rho) [\beta_1, \bullet]_Q \times \sum_{\eta_2} s_{\gamma_2^T/\eta_2}(q^\rho) s_{\eta_2}(q^{\rho+\beta_1}) Q^{|\eta_2|}, \\
C^{(-2,0)}_{\bullet,\bullet,\gamma_2,\gamma_1} &= q^{\kappa(\gamma_2)/2} s_{\gamma_1}(q^\rho) [\gamma_1^T, \bullet]_Q \times \sum_{\eta_2} s_{\gamma_2^T/\eta_2}(q^\rho) s_{\eta_2}(q^{\rho+\gamma_1^T}) Q^{|\eta_2|}.
\end{aligned}
\tag{164}
$$

We sum over $\gamma_2$ when gluing these with subsequent vertices. Note that in these expressions $\beta_1$ is just replaced by $\gamma_1^T$, while the dependence on $\gamma_2$ is the same. Therefore the partition function with brane along $\gamma_1$ is the same as the partition function with brane along $\beta_1^T$.

For a brane on the last vertex the analysis is analogous; there are two possibilities

$$
C^{(-2,0)}_{\bullet,\beta_2,\bullet,\gamma_1} = s_{\beta_2}(q^\rho) [\bullet, \beta_2^T]_Q \times \sum_{\eta_1,\eta_2} s_{\gamma_1/\eta_1}(q^\rho) s_{\eta_1}(q^{\rho+\beta_2^T}) Q^{|\eta_1|},
\tag{165}
$$

$$
C^{(-2,0)}_{\bullet,\bullet,\gamma_2,\gamma_1} = s_{\gamma_2}(q^\rho) [\bullet, \gamma_2]_Q \times \sum_{\eta_1} s_{\gamma_1/\eta_1}(q^\rho) s_{\eta_1}(q^{\rho+\gamma_2}) Q^{|\eta_1|}.
\tag{166}
$$

When gluing with other vertices this expression is summed over $\gamma_1$. Again, the structure of the summand in these two expressions is the same up to replacing $\beta_2^T$ with $\gamma_2$.

**Gluing $(-1,-1)$ geometry**

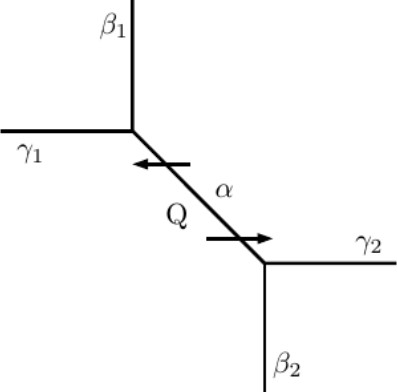

Figure 13: $(-1,-1)$ curve with the inner edge in the canonical framing.

The partition function for the second basic configuration, that is $(-1,-1)$ curve shown in fig. 13, takes form

$$
C^{(-1,-1)}_{\beta_1,\beta_2,\gamma_2,\gamma_1} = \sum_\alpha C^{f_\alpha}_{\alpha\gamma_1\beta_1} C^{f'_\alpha}_{\alpha^T\gamma_2\beta_2} (-1)^{|\alpha|} Q^{|\alpha|}.
\tag{167}
$$

The gluing condition sets $f_\alpha = -f'_\alpha$, with $f_\alpha = v_{\gamma_1} - n v_\alpha$ and $f'_\alpha = v_{\gamma_2} - n' v'_\alpha$. From the geometry we have $v'_\alpha = -v_\alpha$ and $v_{\gamma_2} = -v_{\gamma_1}$, so that the gluing imposes $n = n'$, which yields

$$
C^{(-1,-1)}_{\beta_1,\beta_2,\gamma_2,\gamma_1} = \sum_\alpha C_{\alpha\gamma_1\beta_1} C_{\alpha^T\gamma_2\beta_2}(-1)^{|\alpha|} Q^{|\alpha|} = s_{\beta_1}(q^\rho) s_{\beta_2}(q^\rho)\{\beta_1\beta_2\} \times
$$
$$
\times \sum_{\eta_1,\eta_2} s_{\gamma_1/\eta_1}(q^{\beta_1^T+\rho}) s_{\gamma_2/\eta_2}(q^{\beta_2^T+\rho}) \sum_\kappa s_{\eta_2^T/\kappa^T}(q^{\beta_1+\rho}) s_{\eta_1^T/\kappa}(q^{\beta_2+\rho})(-Q)^{|\eta_1|+|\eta_2|-|\kappa|}.
\tag{168}
$$

We again allow for one brane on a horizontal leg of the first or the last vertex. For the first vertex, upon cyclic permutation of the indices we get

$$
C^{(-1,-1)}_{\beta_1,\beta_2,\gamma_2,\gamma_1} = \sum_\alpha C_{\beta_1\alpha\gamma_1} C_{\alpha^T\gamma_2\beta_2}(-Q)^{|\alpha|} = s_{\gamma_1}(q^\rho) s_{\beta_2}(q^\rho) q^{\kappa(\beta_1)/2} \times
$$
$$
\times \sum_{\eta_1,\eta_2} s_{\beta_1^T/\eta_1}(q^{\gamma_1+\rho}) s_{\gamma_2/\eta_2}(q^{\beta_2^T+\rho}) \sum_\alpha s_{\alpha/\eta_1}(q^{\gamma_1^T+\rho}) s_{\alpha/\eta_2}(q^{\beta_2+\rho})(-Q)^{|\alpha|} q^{-\kappa(\alpha)/2}.
\tag{169}
$$

For a single brane labeled by $\gamma_1$ line, we set $\beta_1 = \beta_2 = \bullet$, which also fixes $\eta_1 = \bullet$. Using

$$
\sum_\alpha s_{\alpha^T/\eta_1}(x) s_{\alpha/\eta_2}(y) = \prod_{i,j}(1+x_i y_j) \sum_\kappa s_{\eta_2^T/\kappa^T}(x) s_{\eta_1^T/\kappa}(y)
\tag{170}
$$

we then find

$$
C^{(-1,-1)}_{\bullet,\bullet,\gamma_2,\gamma_1} = s_{\gamma_1}(q^\rho) \sum_{\eta_2} s_{\gamma_2/\eta_2}(q^\rho) \sum_\alpha s_\alpha(q^{\gamma_1^T+\rho}) s_{\alpha/\eta_2}(q^\rho)(-Q)^{|\alpha|} q^{-\kappa(\alpha)/2} =
$$
$$
= s_{\gamma_1}(q^\rho)\{\gamma_1^T,\bullet\}_Q \sum_{\eta_2} s_{\gamma_2/\eta_2}(q^\rho) s_{\eta_2^T}(q^{\gamma_1^T+\rho})(-Q)^{|\eta_2|}.
\tag{171}
$$

This expression is analogous to the amplitude with a brane on the vertical edge

$$
C^{(-1,-1)}_{\beta_1,\bullet,\gamma_2,\bullet} = s_{\beta_1}(q^\rho)\{\beta_1,\bullet\}_Q \sum_{\eta_2} s_{\gamma_2/\eta_2}(q^\rho) s_{\eta_2^T}(q^{\beta_1+\rho})(-Q)^{||\eta_2|},
\tag{172}
$$

just with $\beta_1$ replaced by $\gamma_1^T$.

Finally, consider a brane labeled by $\gamma_2$, attached to the last vertex. After a cyclic permutation we obtain

$$
C^{(-1,-1)}_{\beta_1,\beta_2,\gamma_2,\gamma_1} = \sum_\alpha C_{\alpha\gamma_1\beta_1} C_{\beta_2\alpha^T\gamma_2}(-Q)^{|\alpha|} = q^{\kappa(\beta_2)/2} s_{\beta_1}(q^\rho) s_{\gamma_2}(q^\rho) \times
$$
$$
\times \sum_{\eta_1,\eta_2} s_{\gamma_1/\eta_1}(q^{\beta_1^T+\rho}) s_{\beta_2^T/\eta_2}(q^{\gamma_1+\rho}) \sum_\alpha q^{\kappa(\alpha)/2} s_{\alpha^T/\eta_1}(q^{\beta_1+\rho}) s_{\alpha^T/\eta_2}(q^{\gamma_2^T+\rho})(-Q)^{|\alpha|}.
\tag{173}
$$

Setting $\beta_1 = \beta_2 = \bullet$ also imposes $\eta_2 = \bullet$, so that

$$
C^{(-1,-1)}_{\bullet,\bullet,\gamma_2,\gamma_1} = s_{\gamma_2}(q^\rho) \sum_{\eta_1} s_{\gamma_1/\eta_1}(q^\rho) \sum_\alpha q^{\kappa(\alpha)/2} s_{\alpha^T/\eta_1}(q^\rho) s_{\alpha^T}(q^{\gamma_2^T+\rho})(-Q)^{|\alpha|} =
$$
$$
= s_{\gamma_2}(q^\rho)\{\bullet,\gamma_2^T\}_Q \sum_{\eta_1} s_{\gamma_1/\eta_1}(q^\rho) s_{\eta_1}(q^{\gamma_2^T+\rho})(-Q)^{|\eta_1|}.
\tag{174}
$$

Compared with the amplitude for a brane on a vertical axis, $\beta_2$ is simply replaced by $\gamma_2^T$

$$
C^{(-1,-1)}_{\bullet,\beta_2,\bullet,\gamma_1} = s_{\beta_2}(q^\rho)\{\bullet,\beta_2\} \sum_{\eta_1,\eta_2} s_{\gamma_1/\eta_1}(q^\rho) s_{\eta_1^T}(q^{\beta_2+\rho})(-Q)^{|\eta_1|}.
\tag{175}
$$

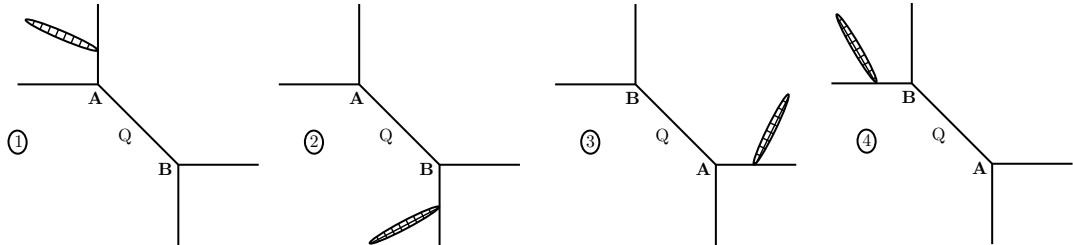

Figure 14: Branes attached to various legs of the conifold diagram.

## A.2 Branes on non-vertical legs

We extend now the above rules by allowing for the brane to be placed on a horizontal leg of the first or the last vertex. Consider placing the brane on the first vertex. The results from the previous section show that in this case the partition corresponding to the brane entering factors $X_{1j}$ is transposed. Consulting then table 2 we see that taking transposed partition is equivalent to switching all vertices types to the opposite. This is then consistent with our convention for choosing the type of the first vertex. Indeed changing position of the brane leads then to the opposite vertex type.

For a brane attached to the last vertex we adopt the following convention. We choose the type of the first vertex as if the brane was there on the vertical leg. This fixes types of all vertices. Specifically it fixes the type of the last vertex for the case when the brane is on the last vertex on the vertical line. On the other hand, if we move the brane to the horizontal line, all vertices change their types to opposite ones. This then leads to the prescription given in the paragraph above table 1.

## A.3 Explicit computations for the conifold and the resolution of $\mathbb{C}^3/\mathbb{Z}_2$

Let us illustrate the formalism presented above in examples of the conifold and the resolution of $\mathbb{C}^3/\mathbb{Z}_2$. Consider branes in positions 1-2-3-4, as shown respectively in fig. 14 and 15. In these cases, contributions to the open string partition function take the following form:

| Position | Conifold | Resolved $\mathbb{C}^3/\mathbb{Z}_2$ |
|---|---|---|
| 1 | $(\underline{A},B) \to s_{\beta_1}(q^\rho)\{\beta_1,\bullet\}_Q$ | $(\underline{A},A) \to s_{\beta_1}(q^\rho)[\beta_1,\bullet]_Q$ |
| 2 | $(A,\underline{B}) \to s_{\beta_2}(q^\rho)\{\bullet,\beta_2\}_Q$ | $(A,\underline{A}) \to s_{\beta_2}(q^\rho)[\bullet,\beta_2^T]_Q$ |
| 3 | $(B,\underline{A}) \to s_{\gamma_2}(q^\rho)\{\bullet,\gamma_2^T\}_Q$ | $(B,\underline{B}) \to s_{\gamma_2}(q^\rho)[\bullet,\gamma_2]_Q$ |
| 4 | $(\underline{B},A) \to s_{\gamma_1}(q^\rho)\{\gamma_1^T,\bullet\}_Q$ | $(\underline{B},B) \to s_{\gamma_1}(q^\rho)[\gamma_1^T,\bullet]_Q$ |

For example, the partition function for a single brane in framing $f$ at position 1 reads

$$\psi_1(x) = \sum_{n=0}^\infty ((-1)^n q^{n(n-1)/2})^{f+1} \frac{x^n}{(q;q)_n} (Q;q)_n. \tag{176}$$

# Acknowledgements

We thank Andrea Brini for discussions on these and related topics. The work of TK has been supported in part by "Investissements d'Avenir" program, Project ISITE-BFC (No. ANR-15-IDEX-0003), and EIPHI Graduate School (No. ANR-17-EURE-0002). The work of MP has been

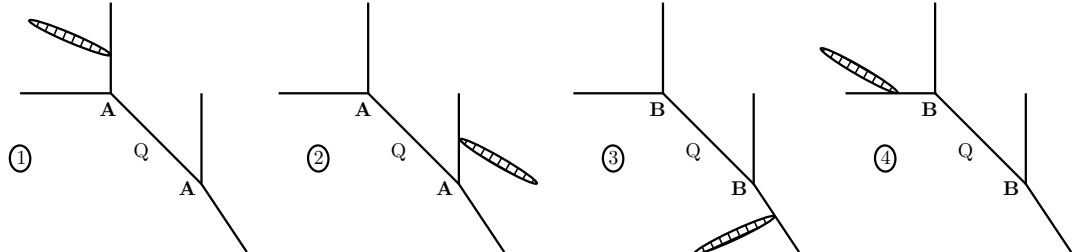

Figure 15: Branes attached to various legs of a diagram for the resolution of $\mathbb{C}^3/\mathbb{Z}_2$.

supported by the National Science Centre, Poland, under the SONATA grant 2018/31/D/ST3/03588. The work of YS has been supported by the national Natural Science Foundation of China (Grants No.11675167 and No.11947301). The work of PS has been supported by the TEAM programme of the Foundation for Polish Science co-financed by the European Union under the European Regional Development Fund (POIR.04.04.00-00-5C55/17-00).

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
