# Peer review of "Branes, quivers and wave-functions"

_SciPost Physics, doi:SciPost Phys. 10, 051 (2021)_

## Round 1 · Referee Report · Anonymous (Referee 1) · 2021-1-12

Report

The knots-quiver correspondence (P. Kucharski, M. Reineke, M. Stosic and P. Sulkowski, BPS states, knots and quivers, Phys. Rev. D 96, 121902 (2017)) states that several properties of the topological string on a toric threefold with a brane configuration can be encoded in a quiver, an oriented graph. Quiver representation theory is a very rich and important branch of algebra, with several applications in physics. In string theory this setup can be used to engineer a knot, for example with a brane configuration on the resolved conifold. Then the topological partition functions, which correspond to certain polynomials associated with the knot, can be recast as a quiver generating series.

The paper expands the scope of this correspondence in a different setup, namely strip geometries, a class of toric threefolds without compact 4-cycles, with the insertion of a basic Aganagic-Vafa lagrangian brane. In this case the brane partition functions behave as wave-functions in different polarizations. The question addressed in the paper is how is this behaviour compatible with the quiver description. The authors determine operations on quivers which reproduce the effect of moving the branes or doing an $SL(2,Z)$ transformation, and discuss the consequences of these properties.

For a given brane in a toric strip geometry the open topological string partition function can be computed using the formalism of the topological vertex. These computations are reviewed in Section 2 of the paper (and the formalism slightly extended in Appendix A). The resulting wave function $\psi (x)$ is annihilated by an operator $\hat{A} (\hat{x} , \hat{y})$, the so-called quantum curve (where $\hat{x}$ acts by multiplication and $\hat{y}$ by $\hat{y} f(x) = f (q x)$). The name originates from the fact that the $q \longrightarrow 1$ limit of this operator defines the classical mirror curve.

One of the main results of the paper is that this brane wave function can be re-written as a motivic generating series for a symmetric quiver. Such motivic series arise in the theory of Donaldson-Thomas invariants; they are many variable series which contain information about the moduli spaces of quiver representations and which can be decomposed uniquely as a product of quantum dilogarithms whose exponents are Donaldson-Thomas invariants. This connection between physics, representation theory and Donaldson-Thomas theory is one of the most interesting aspects of the knots-quiver correspondence. The authors extend the same arguments also to the case where the strip geometry is periodic where two opposite edges of the toric diagram of the strip, in the first and last vertex, are identified.

The topological string partition function for branes, on toric manifolds, is conjectured to behave as a wave function. A change in polarization corresponds to changing the position of a single brane on a fixed toric manifold from, say the $i^{th}$ to the $j^{th}$ vertex in the strip. This transformation is implemented via an integral transform determined by an element of $SL(2;Z)$. The authors check explicitly that only a subset of $SL(2;Z)$, namely those transformations generated by $T$ and $S^2$ lead to wave-functions which can be written in quiver form. Physically these are the transformations which correspond to changing the location of a brane. The $S$ transformation makes an appearance only in special situations, such as the conifold where the A-polynomial is symmetric. The analysis of the integral kernel which determines the transformations is done at the semi-classical level; the authors conjecture it should hold at the full quantum level too.

This behaviour is traced back to the action of $SL(2;Z)$ on the A-polynomial in Section 4. There the authors show by direct computations that the A-polynomial is preserved only by the $S^2$ and $T$ operations generically, and by $S$ in special cases (affine space and resolved conifold). The authors also discuss the effect of these operations on the physical setup, as moving a brane from one vertex to another or within the same vertex.

The behaviour of the brane partition function described above, is finally checked explicitly in a few examples in Section 5.

The paper is well written and clear, and contains enough details to follow the computations (although I haven't gone through all of them). The results are a nice confirmation of certain aspects of the conjectural knot-quiver correspondence. They are important since they provide further evidence for the correspondence and help to sharpen it. I therefore recommend the paper for publication.

---

## Round 1 · Referee Report · Anonymous (Referee 2) · 2021-2-15

Report

The paper by Kimura, Panfil, Sugimoto and Sulkowski considers partition functions of branes on a class of toric varieties called strip geometries (and their periodic generalization), from three perspectives: that of the open topological string, of the A-polynomial, and of quiver representations, generalizing previous work on the geometries $\mathbb{C}^3$ and the conifold.

The computations of the paper are anchored in the topological vertex formalism, which permits the computation of the open topological string partition function on toric geometries in the presence of Lagrangian branes. From this vantage point, the authors derive the A-polynomials for such geometries, as well as, notably, the associated quivers. The former are difference operators which annihilate the partition function; their $q \rightarrow 1$ limit coincides with the mirror curve of the underlying toric geometry. Given the partition function, they are straightforward to derive. The latter give rise to so-called motivic generating series; the authors identify the quiver associated to a toric brane configuration by matching this series to the partition function.

Having studied the behavior of the partition function, the A-polynomial, and the associated quiver representation upon changing the position of the Lagrangian brane, the authors proceed to investigate how this operation can be encoded in terms of an integration of the partition function against an integration kernel. This is motivated by previous work in the literature, in which the brane partition function is interpreted as a wave-function, and the change of brane position as a canonical transformation on the underlying system. Evaluating such integrals in the semi-classical limit (and permitting themselves a rescaling of the Kaehler parameters), the authors conclude that only a subgroup of the group of canonical transformations, generated by the operators $T$ and $S^2$, correspond to the operation of moving branes. Finally, the authors address the same question from the vantage point of the A-polynomial.

The paper concludes by working out numerous examples in detail.

This paper advances the investigation of an as yet somewhat mysterious relation, that between the open topological string partition function, its wave function interpretation, and the nature of the difference operator that annihilates it, on the computational front. It is clearly written and calculations are well presented. It contains novel results which should help in uncovering the underlying structures. I therefore recommend publication.

---

## Editorial Decision

published